# A protein-independent fluorescent RNA aptamer reporter system for plant genetic engineering

Jiuyuan Bai[1,5], Yao Luo [2,5], Xin Wang[1,5], Shi Li[1], Mei Luo[1], Meng Yin[1], Yuanli Zuo[1], Guolin Li[1], Junyu Yao[1], Hua Yang[1], Mingdi Zhang[1], Wei Wei[1], Maolin Wang[1], Rui Wang [1✉], Chunhai Fan [3,4✉] & Yun Zhao [1✉]

Reporter systems are routinely used in plant genetic engineering and functional genomics research. Most such plant reporter systems cause accumulation of foreign proteins. Here, we demonstrate a protein-independent reporter system, 3WJ-4 × Bro, based on a fluorescent RNA aptamer. Via transient expression assays in both *Escherichia coli* and *Nicotiana benthamiana*, we show that 3WJ-4 × Bro is suitable for transgene identification and as an mRNA reporter for expression pattern analysis. Following stable transformation in *Arabidopsis thaliana*, 3WJ-4 × Bro co-segregates and co-expresses with target transcripts and is stably inherited through multiple generations. Further, 3WJ-4 × Bro can be used to visualize virus-mediated RNA delivery in plants. This study demonstrates a protein-independent reporter system that can be used for transgene identification and in vivo dynamic analysis of mRNA.

[1] Key Laboratory of Bio-Resource and Eco-Environment of Ministry of Education, College of Life Sciences, Sichuan University, Chengdu 610065, China. [2] State Key Laboratory and Collaborative Innovation Center of Biotherapy, West China Hospital, Sichuan University, Chengdu 610041 Sichuan, China. [3] School of Chemistry and Chemical Engineering, Frontiers Science Center for Transformative Molecules, Institute of Translational Medicine, Shanghai Jiao Tong University, Shanghai 200240, China. [4] Institute of Molecular Medicine, Shanghai Key Laboratory for Nucleic Acids Chemistry and Nanomedicine, Renji Hospital, School of Medicine, Shanghai Jiao Tong University, Shanghai 200127, China. [5]These authors contributed equally: Jiuyuan Bai, Yao Luo, Xin Wang. ✉email: wangray1987@scu.edu.cn; fanchunhai@sjtu.edu.cn; zhaoyun@scu.edu.cn

Plant genetic manipulation techniques are powerful tools in functional genomics research and crop genetic improvement. Using these techniques, target genes can be transformed into recipient plants to achieve various goals, such as gain or loss-of-function[1]. Marker genes, which are critical to efficient transgene plant development, fall into two categories: selectable marker genes and reporter genes[2]. Selectable marker genes, such as antibiotic resistance and herbicide resistance genes, are used to enrich positive transformed cells based on their ability to confer resistance to toxic substances[3].

Reporter genes do not provide a selective advantage to cells but rather are used to confirm transgenic events because they allow visual detection of transformed cells and tissues. Although numerous reporter genes, including those encoding fluorescent proteins (FPs), β-glucuronidase (GUS), and luciferase (Luc)[4], have been employed to identify transgenic lines or visualize target gene expression patterns in plants, they all have limitations[4,5]. For example, the green fluorescent protein gene (GFP) is commonly employed to track subcellular localization of proteins[6]; however, the high autofluorescence of GFP in plant organs often makes it inappropriate for large-scale expression analysis at the whole-plant level[7]. The *GUS* gene is widely used for tissue-specific expression analysis of target proteins because GUS is readily detectable by histochemical staining[8], but it cannot be used to measure the dynamic expression of target genes because of the destructive histochemical staining and destaining procedure involved in its detection[9]. The Luc gene has also been frequently used to monitor real-time gene expression in plant[10–12]; however, when measured based on bioluminescence, the Luc activity can be easily affected by substrate availability or inherent differences in local cell environments, therefore making the Luc gene unsatisfactory for accurately monitoring tissue- or cell-specific expression patterns[13].

Although these reporters can be used for selection of multiple generations of transgenic plants, these systems do not directly detect the presence of the target gene in progeny generations, in which the partial integration of T-DNA may occur[14]; moreover, the expression levels of target genes are not directly reflected by those of the reporter genes. Therefore, PCR and real-time quantitative PCR (qRT-PCR), as well as northern blotting, must be used to reconfirm the status of transgenic lines, which is laborious and time consuming. Conversely, a common approach to characterize the expression pattern of a target gene is to use the gene's native promoter to drive the expression of a reporter gene, but this does not accurately reflect the normal target gene expression pattern because some expression regulatory elements may occur outside of the cloned promoter region[15,16]. In addition, such reporter systems are based on protein products, which can negatively affect gene expression and increase energy utilization in transgenic plants due to the large amount of foreign protein accumulation in cells[17]. Moreover, translational fusions cannot be used to report the expression of non-coding RNA. Consequently, there is a need for a reporter system to directly monitor the expression of different categories of target genes without the accumulation of foreign proteins.

To accomplish this, we focused on reporter systems acting at the transcriptional level. An RNA-based reporter system approach could circumvent some of the limitations of protein-dependent reporter systems. In creating such systems, the greatest challenge is that of exploiting unique markers to allow dynamic imaging of RNAs in living plant cells and tissues. Reif et al. fused the malachite green aptamer, the ribozyme and a siRNA to the bacteriophage phi29 packaging RNA three-way junction (3WJ) motif to generate RNA nanoparticles, which was demonstrated an excellent tool for monitoring RNA folding and degradation in real time in living cells[18]. Besides, several RNA aptamers, with names such as Spinach and Broccoli, have been also created for live-cell imaging of RNA based on a small fluorophore whose fluorescence is activated upon binding and sequestration within the aptamer[19,20]. Spinach and Broccoli both produce green fluorescence, but Broccoli is superior to Spinach due to the shorter sequence, brighter fluorescence, and higher affinity to fluorophore, (Z)-4-(3,5-difluoro-4-hydroxybenzylidene)-2-methyl-1-(2,2,2-trifluoroethyl)-1H-imidazol -5(4H)-one (DFHBI-1T)[21]. Recently, several aptamers such as Pepper[22] and Mango II[23], based on other fluorophores, have been reported for live-cell imaging of RNA. These reports showed RNA aptamers possess application prospect for RNA imaging in transiently transformed *Escherichia coli* and mammalian cells[24–27]. Huang et al. designed a Spinach aptamer RNA-based sensor, which is successfully used for quantification and localization of small RNAs in mammalian cells. They also tried to express this RNA aptamer in protoplasts and *Nicotiana benthamiana* leaves but no reliable signal was detected[28]. So far, there is no report of any aptamer that can be used for RNA imaging in live plant cells. Therefore, we set out to create fluorescent aptamers to apply the RNA aptamer approach to plants, aiming to develop a reporter system for plant genetic engineering at the transcriptional level.

In this study, we design a series of fluorescence aptamers based on the modified 3WJ scaffold and the optimized Broccoli sequence (Bro). We investigate the performance of the 3WJ-nBro series and the previously developed F30-Bro in RNA imaging both in vitro and in vivo. 3WJ-4 × Broccoli (3WJ-4 × Bro) shows bright fluorescence in *N. benthamiana* cells, whereas neither F30-Bro nor the other aptamers we created produce a detectable signal, revealing 3WJ-4 × Bro to be a satisfactory aptamer for mRNA imaging in plant cells. Next, we use 3WJ-4 × Bro as a gene marker in creating a system for reporting the presence and expression of target genes at the transcriptional level in plants through fluorescence visualization. Our results show that 3WJ-4 × Bro can report the transient expression of target genes in plant cells at the transcriptional level, obviating the need for accumulation of foreign proteins. Furthermore, we establish that 3WJ-4 × Bro is a useful marker for the identification of stably transformed *Arabidopsis thaliana* and for expression analysis of target genes in different tissues but does not perturb the processes of transcription and translation. Notably, 3WJ-4 × Bro can be co-segregated and co-expressed with target genes and is then stably inherited through multiple generations of transgenic plants in a Mendelian fashion. In addition, we are able to directly detect the process of RNA delivery in *N. benthamiana* using our reporter system combined with a modified *N. benthamiana* rattle virus (TRV) system. In summary, we report here the development of a protein-independent reporter system for plant genetic engineering, which should facilitate both functional genomics research and the genetic improvement of crops.

## Results

**Construction and screening of RNA fluorescence aptamer**. Starting from the previously reported F30-Broccoli[2] (F30-Bro), we modified the F30 structure to form a 3WJ and optimized the Broccoli sequence. To screen for an optimum aptamer with strong fluorescence and high stability, we created a series of 3WJ-nBro aptamers, 3WJ-Broccoli (3WJ-Bro), 3WJ-2 × Broccoli (3WJ-2 × Bro), 3WJ-4 × Bro, and 3WJ-6 × Broccoli (3WJ-6 × Bro), based on the modified 3WJ scaffold and two linkers (Supplementary Fig. 1). We then in vitro transcribed each member of the 3WJ-nBro series, as well as the original F30-Bro, individually with T7 RNA polymerase and incubated them with 10 μM DFHBI-1T. The spectra of all the 3WJ-nBro series had the same single emission peak at 527 nm and the same single excitation

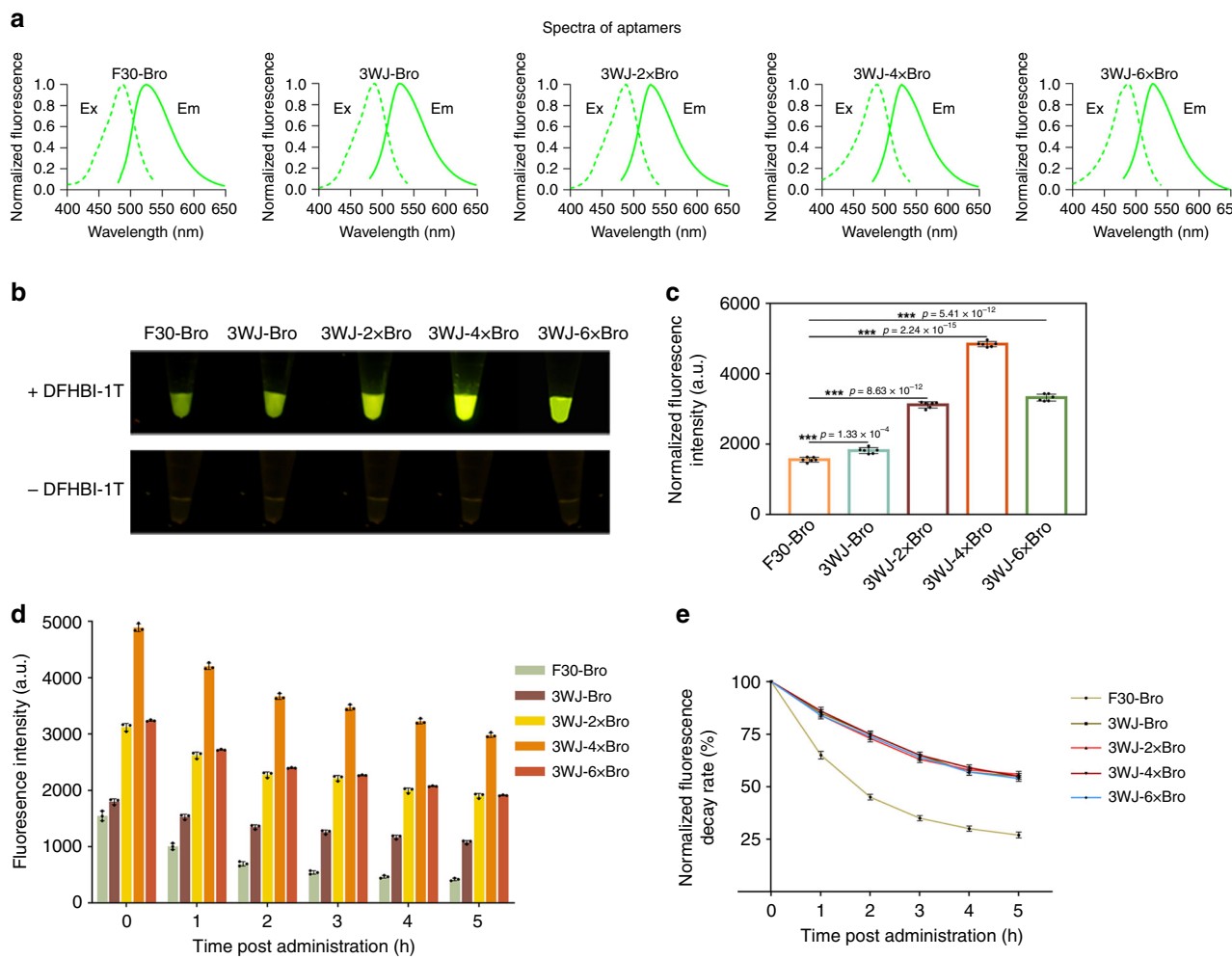

**Fig. 1 In vitro performance of the 3WJ-nBro series. a** Excitation (dashed) and emission (solid) spectra of RNA aptamers. **b** Representative fluorescence images of 3WJ-nBro series from a Blue Light Gel Imager (three replicated trials). **c** Fluorescence quantification of 3WJ-nBro series. Error bars depict the mean ± standard deviation (s.d.) ($n = 6$) and $p$ values are calculated using an unpaired, two-tailed, Student's $t$ test. ***$p < 0.001$. **d** Fluorescence detection within 5 h after DFHBI-1T administration ($n = 3$). **e** Normalized fluorescence decay rate calculated as the ratio of remaining fluorescence to initial fluorescence ($n = 3$). All experiments above were repeated three independent times. Source Data are provided as a Source Data file.

**Table 1 The spectral characteristics of fluorescent RNA aptamers.**

| Aptamers | Optimal excitation (nm) | Optimal emission (nm) | Extinction coefficient ($M^{-1} cm^{-1}$) |
|---|---|---|---|
| F30-Bro | 485 | 525 | 14,580 |
| 3WJ-Bro | 488 | 527 | 19,490 |
| 3WJ-2 × Bro | 488 | 527 | 31,870 |
| 3WJ-4 × Bro | 488 | 527 | 48,390 |
| 3WJ-6 × Bro | 489 | 527 | 33,420 |

Extinction coefficients were all measured at pH 7.4.

peak at 488 nm, similar to the spectrum of the F30-Bro (Fig. 1a and Table 1). The extinction coefficient of all the 3WJ-nBro series was increased compared with F30-Bro, among which the extinction coefficient of 3WJ-4 × Bro was the highest, reaching 48390 $M^{-1} cm^{-1}$ (2.32-fold higher than that of F30-Bro) (Table 1). All of the aptamer RNAs produced green fluorescence signals after excitation with 488-nm light, and the 3WJ-nBro series exhibited higher brightness than F30-Bro (Fig. 1b). Quantitative fluorescence data showed significantly higher

fluorescence intensity for 3WJ-Bro compared with F30-Bro, indicating that our 3WJ scaffold improved the fluorescence intensity of Broccoli (Fig. 1c). Furthermore, the fluorescence intensities of 3WJ-2 × Bro and 3WJ-4 × Bro were 0.5-fold and 1.2-fold higher, respectively, than that of 3WJ-Bro, whereas that of 3WJ-6 × Bro was lower than that of 3WJ-4 × Bro but 0.7-fold higher than that of 3WJ-Bro. Thus, embedding multiple tandem-repeated Broccoli units in the 3WJ scaffold and linker did promote fluorescence intensity, and embedding four Broccoli tandem repeats in the scaffold was optimal (Fig. 1b, c). To compare the stability of the four 3WJ-nBro series and F30-Bro, we measured the fluorescence of each aptamer–DFHBI-1T complex at 488-nm excitation every hour for 5 h. The remaining fluorescence was higher for 3WJ-nBro series than for F3-Bro throughout the 5 h, and the fluorescence of 3WJ-4 × Bro was still 2981.68 a.u. after 5 h, which was sixfold and twofold higher than those of F30-Bro and 3WJ-Bro, respectively (Fig. 1d). We also calculated the fluorescence decay rate after normalization to the corresponding initial fluorescence intensity. F30-Bro showed quicker fluorescence decay than the 3WJ-nBro series: after 5 h, 65% of the initial fluorescence remained for 3WJ-4 × Bro, but only 27% for F30-Bro (Fig. 1e). However, no significant difference of fluorescence decay rate was detected among the 3WJ-nBro series (Fig. 1e). Thus, all

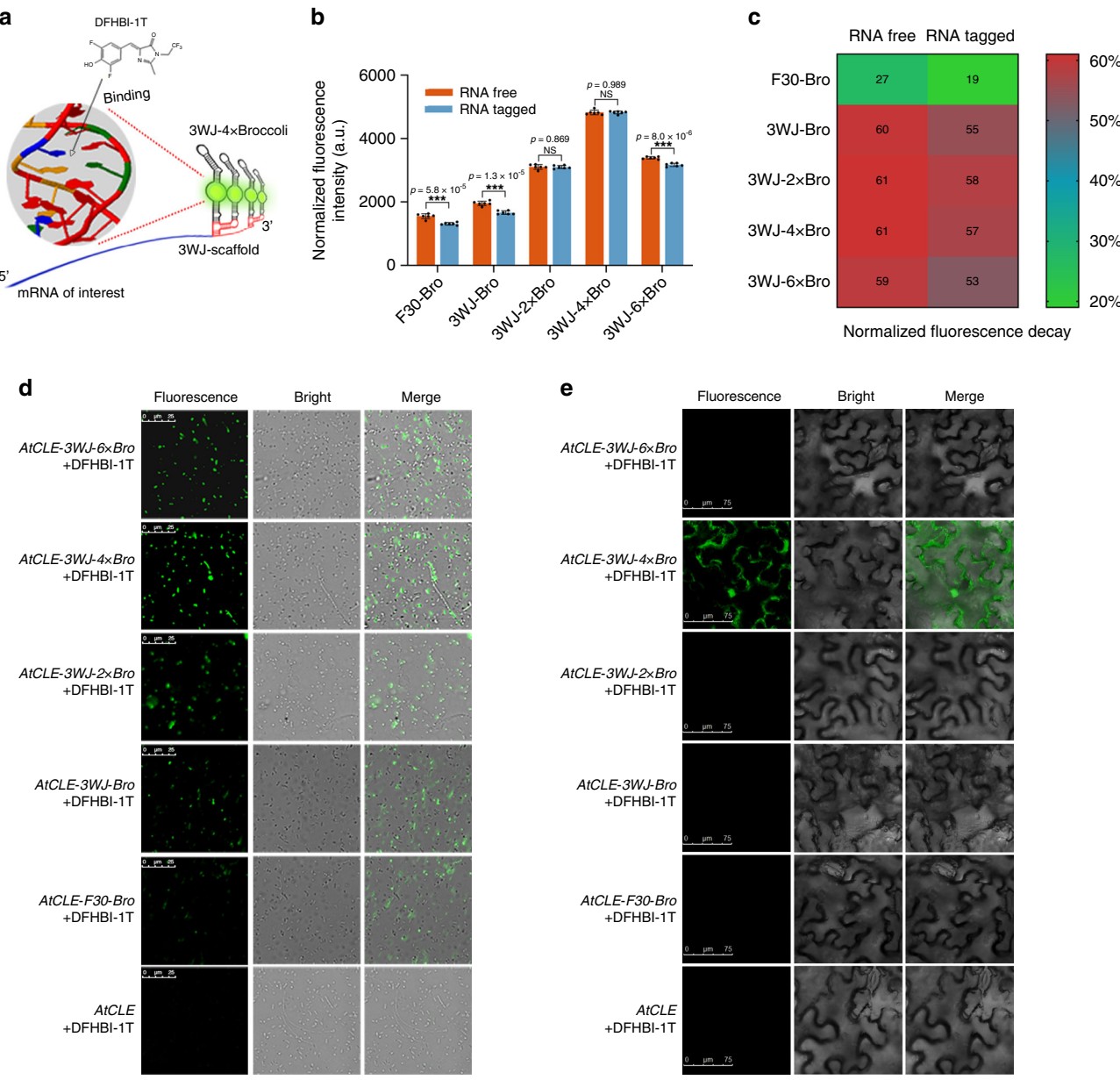

**Fig. 2 Comparison of 3WJ-nBro series for imaging mRNA in vitro and in vivo. a** Schematic view of mRNA imaging by 3WJ-nBro series. **b** Fluorescence changes of 3WJ-nBro series after appending to the 3′ end of *AtCLE* mRNA. Error bars depict the mean ± s.d. ($n = 6$) and $p$ values are calculated using an unpaired, two-tailed, Student's $t$ test. \*\*\*$p < 0.001$. NS, for not significant. **c** Differential fluorescence decay of 3WJ-nBro series with and without *AtCLE* mRNA ($n = 6$). **d** Representative fluorescence images of 500 *E. coli* BL21 cells expressing 3WJ-nBro-tagged *AtCLE* mRNAs after incubation with 10 μM DFHBI-1T. Scale bars, 25 μm. **e** Fluorescence images of *N. benthamiana* leaf cells expressing 3WJ-nBro-tagged *AtCLE* mRNAs after incubation with 10 μM DFHBI-1T. Scale bars, 75 μm. All experiments above were repeated three independent times. Source Data underlying **b**–**e** are provided as a Source Data file.

four 3WJ-nBro series produce more stable fluorescence than F30-Bro, and 3WJ-4 × Bro produces the most.

To assess whether the 3WJ-nBro series could be used to detect mRNA in vitro, we appended each 3WJ-nBro and F30-Bro downstream of the terminal codon of the target gene *AtCLE* as a means to transcribe the fusion mRNA in vitro. The target mRNA could be detected from the lighted-up fluorescence of aptamers after DFHBI-1T binding (Fig. 2a). We used the RNAfold web server to predict the minimum free energy (MFE) secondary structures of the five aptamer-tagged mRNAs, which suggested that these aptamers could keep their original MFE structures after being used to tag mRNAs (Supplementary Fig. 2). However, fluorescence imaging and quantification of the fusion mRNAs indicated that F30-Bro, 3WJ-Bro, and 3WJ-6 × Bro all had

significantly decreased fluorescence intensity when attached to *AtCLE* mRNA, whereas 3WJ-2 × Bro and 3WJ-4 × Bro showed no significant effects on fluorescence intensity upon mRNA attachment (Fig. 2b and Supplementary Fig. 3). Comparison of the fluorescence intensities among the fusion mRNAs indicated that *AtCLE* mRNA tagged with 3WJ-4 × Bro (*AtCLE-3WJ-4 × Bro*) produced the strongest fluorescence signal, of up to 4300 a.u., followed by *AtCLE-3WJ-2 × Bro*, *AtCLE-3WJ-6 × Bro*, *AtCLE-3WJ-Bro*, and *AtCLE-F30-Bro* (Fig. 2b). We also assessed the stability of the fluorescence of free and mRNA-fused aptamers, as evidenced by fluorescence decay over 5 h. F30-Bro, 3WJ-Bro, and 3WJ-6 × Bro had distinctly lower fluorescence stability when appended to the 3′ terminus of *AtCLE* mRNA, whereas 3WJ-2 × Bro and 3WJ-4 × Bro showed similar fluorescence

stability before and after fusion to the mRNA (Fig. 2c). Consequently, considering both fluorescence intensity and stability, 3WJ-4×Bro showed the greatest potential for use in RNA imaging in vitro among the five aptamers tested.

To explore the performance of 3WJ-nBro series in RNA imaging in vivo, we constructed a series of constructs expressing 3WJ-nBro-tagged *AtCLE* mRNAs and tested them in both *E. coli* and *N. benthamiana* cells. Confocal microscopy imaging showed that more than 50% of *E. coli* cells expressing *AtCLE-3WJ-2×Bro*, *AtCLE-3WJ-4×Bro*, or *AtCLE-3WJ-6×Bro* fluoresced, whereas <30% of cells expressing *AtCLE-3WJ-Bro* or *AtCLE-F30-Bro* showed detectable fluorescence (Fig. 2d and Supplementary Fig. 4). In addition, the cells expressing *AtCLE-F30-Bro* showed considerably weaker average fluorescence than those expressing any of *AtCLE-3WJ-nBro* mRNAs (Fig. 2d and Supplementary Fig. 5), and cells expressing *AtCLE-3WJ-4×Bro* showed the highest fluorescence. Moreover, in *N. benthamiana* cells, only cells expressing *AtCLE-3WJ-4×Bro* mRNA exhibited bright fluorescence under the confocal microscope, whereas cells expressing any of the other aptamer-tagged mRNAs showed no detectable fluorescence (Fig. 2e). These results indicated that 3WJ-4×Bro was the only fully competent aptamer that could be successfully used for mRNA imaging in both live *E. coli* cells and plant cells. Consequently, we used 3WJ-4×Bro in constructing our transcriptional-level reporter system.

**Characteristics of 3WJ-4×Bro as an in vitro mRNA reporter**. The detection of most endogenous RNAs requires highly sensitive imaging probes due to their low expression level. To confirm the detectable level of the 3WJ-4×Bro marker, we designed a series of concentrations for in vitro transcripts of the 3WJ-4×Bro-tagged *AtCLE* (*AtCLE-3WJ-4×Bro*), incubated the dilutions with 10 μM DFHBI-1T in 50 μL HEPES, and imaged them. Fluorescence signals could be observed when the transcript content was as low as 9.375 nmol, indicating that 3WJ-4×Bro has high detection sensitivity in mRNA imaging (Supplementary Fig. 6). Furthermore, linear regression showed that the fluorescence intensity increased linearly with increasing transcript contents from 9.375 to 300 nmol (Fig. 3a).

To determine whether 3WJ-4×Bro was a useful marker for reporting the expression of different target genes, we tested three genes of different sequence lengths: *AtCLE*, *mCherry*, and *NtTubα*. We fused 3WJ-4×Bro to the 3′-untranslated region (3′-UTR) of each test gene to create a chimeric gene series (*AtCLE-3WJ-4×Bro*, *mCherry-3WJ-4×Bro*, and *NtTubα-3WJ-4×Bro*). In vitro imaging of equimolar amounts of transcripts of the three genes showed similar fluorescence levels for each (Fig. 3b), and quantification data showed average fluorescence intensities for each 3WJ-4×Bro-tagged mRNA of more than 4000 a.u., with no significant difference among the three (Fig. 3c), suggesting that 3WJ-4×Bro is sufficiently stable to avoid interference from the different target mRNAs. Results from a fluorescence decay assay showed that 3WJ-4×Bro had greater fluorescence stability after attachment to the 3′-UTR of each of the three mRNAs for 5 h after adding DFHBI-1T. In addition, we observed an increased fluorescence stability for 3WJ-4×Bro with increasing length of the mRNA sequence over 5 h, indicating that mRNAs with higher molecular weights may enhance the stability of the 3WJ-4×Bro tag (Fig. 3d). These favorable detection sensitivity and stability characteristics make 3WJ-4×Bro a noteworthy potential marker for tracking different mRNAs.

**Performance of 3WJ-4×Bro in a prokaryotic expression system**. To explore the capability of 3WJ-4×Bro as a marker for reporting functional gene expression in prokaryotic cells, we

transformed the constructs harboring the three chimeric genes (*AtCLE-3WJ-4×Bro*, *mCherry-3WJ-4×Bro*, and *NtTubα-3WJ-4×Bro*) into *E. coli* BL21 cells (Supplementary Fig. 7). Confocal imaging showed robust green fluorescence in *E. coli* cells expressing each of the three 3WJ-4×Bro-tagged mRNAs. As a comparison, cells containing either untagged mRNA or DFHBI-1T alone showed no detectable fluorescence signal (Fig. 4a and Supplementary Fig. 8). We also assessed the detection efficiency based on the ratio of fluorescent cells to total cells. Approximately 80% of cells expressing each of the three mRNAs tagged with 3WJ-4×Bro fluoresced, indicative of high detection efficiency for 3WJ-4×Bro in reporting the transcription of different target genes in *E. coli* cells (Fig. 4b). In addition, quantitation of the fluorescence of cells expressing the three 3WJ-4×Bro-tagged mRNAs showed similar fluorescence intensities in all three, suggesting that the 3WJ-4×Bro marker did not perturb mRNA transcription of any of the three target genes in living *E. coli* cells (Fig. 4c).

Most heterologous RNA aptamers are cleaved by RNase in live cells, causing them to dissociate from the target RNAs. To ensure that the fluorescence observed in *E. coli* precisely reflected the distribution of the mRNA, rather than free 3WJ-4×Bro, we conducted an integrity analysis of the 3WJ-4×Bro-tagged mRNAs by in-gel imaging of fluorescent RNAs, in which total cellular RNAs from induced *E. coli* cells were separated in a urea denaturing gel and stained with DFHBI-1T. The gel image showed a single band of the expected size for each sample (Fig. 4d). Furthermore, when we excised the RNA bands, eluted them from the gel, and subjected them to reverse transcription sequencing, the cDNA sequence corresponded to the full-length sequence of each 3WJ-4×Bro-tagged mRNA (Supplementary Fig. 9). These results confirmed that the fluorescence signal seen in *E. coli* cells came only from the intact 3WJ-4×Bro-tagged mRNAs, indicating that 3WJ-4×Bro detection accurately reports the expression level of functional genes in prokaryotic cells.

In addition, to explore the effect of 3WJ-4×Bro on the translation activity of target mRNAs, we transformed the different test genes, with and without the 3WJ-4×Bro marker, into *E. coli* strain Rosetta (DE3) cells for protein expression analysis. The results of sodium dodecyl sulfate–polyacrylamide gel electrophoresis (SDS-PAGE) and Coomassie blue staining showed that cells harboring the test genes both with and without the 3WJ-4×Bro marker expressed target proteins of the expected size and showed the same distribution of total protein on the gel (Supplementary Fig. 10). For further verification, we performed immunoblotting. The target proteins expressed from 3WJ-4×Bro-tagged mRNAs had the same molecular weights and expression levels as those from untagged mRNAs (Fig. 4e). These data confirmed 3WJ-4×Bro as an excellent gene marker for reporting the expression of different target genes in prokaryotic systems that does not perturb either transcription or translation.

**Performance of 3WJ-4×Bro in plant transient expression system**. The spatiotemporal expression and dynamics of mRNAs play critical roles in directly and indirectly regulating complex biological processes in plants. To further explore the utility of 3WJ-4×Bro as a genetically encoded marker in excised plant cells, we conducted a protoplast transient expression assay. We expressed each of the three 3WJ-4×Bro-tagged mRNAs (*AtCLE-3WJ-4×Bro*, *mCherry-3WJ-4×Bro*, and *NtTubα-3WJ-4×Bro*) in protoplasts under the control of the CaMV *35S* promoter (Supplementary Fig. 11a). Confocal imaging of the protoplasts showed bright fluorescence in those expressing 3WJ-4×Bro-tagged mRNAs after incubation with 10 μM DFHBI-1T but no signal in the absence of either 3WJ-4×Bro or DFHBI-1T (Fig. 5a and

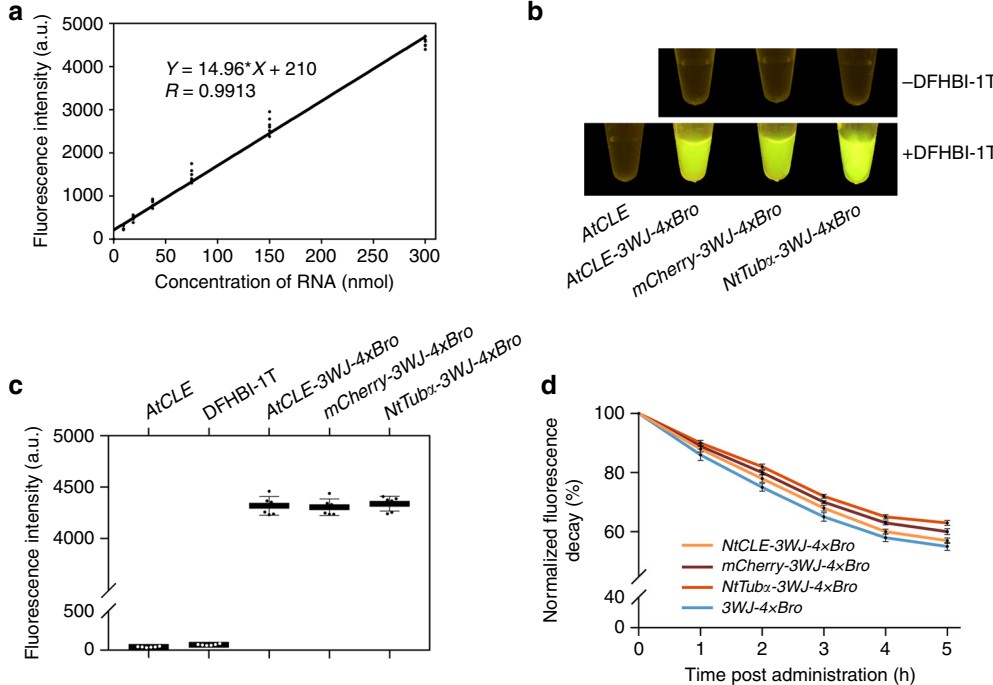

**Fig. 3 Characteristics of 3WJ-4 × Bro after tagging various mRNAs in vitro. a** Linear regression of fluorescence intensity and mRNA content ($n = 6$). **b** Representative image of 300 μM 3WJ-4 × Bro-tagged mRNAs in vitro after incubation with 10 μM DFHBI-1T. **c** Quantification of the fluorescence of RNA–DFHBI-1T complexes in **b** using a fluorescence spectrophotometer. Error bars, mean ± s.d. ($n = 6$). **d** Comparison of the fluorescence decay of 3WJ-4 × Bro-tagged mRNAs within 5 h after DFHBI-1T administration. The fluorescence decay was calculated after normalization of the remaining fluorescence intensity to the respective initial fluorescence intensity. Error bars, mean ± s.d. ($n = 6$). All experiments were repeated three independent times. Source Data are provided as a Source Data file.

Supplementary Fig. 11b), indicating that 3WJ-4 × Bro could be used to report gene expression in plant cells based on the green fluorescence signal. We also observed a distinct cytoplasmic localization of the fluorescence signal for all three 3WJ-4 × Bro-tagged mRNAs (Fig. 5a). Extracellular and intracellular fluorescence detection revealed a high 3WJ-4 × Bro fluorescence signal but only low DFHBI-1T background fluorescence (Supplementary Fig. 11c). All three 3WJ-4 × Bro-tagged mRNAs showed a high signal-to-background ratio (S/B) in the green fluorescence channel (Fig. 5b), and quantification showed similar expression levels for all three target genes in protoplasts (Supplementary Fig. 12). Taken together, these results define 3WJ-4 × Bro as a favorable marker for reporting mRNA expression in excised plant cells.

To further assess the performance of 3WJ-4 × Bro in tracking target gene expression in plant tissues, we introduced the chimeric genes (*AtCLE-3WJ-4 × Bro*, *mCherry-3WJ-4 × Bro*, and *NtTubα-3WJ-4 × Bro*) into *N. benthamiana* leaf tissue by *Agrobacterium*-mediated transformation for transient expression. Confocal imaging showed bright fluorescence in the leaves expressing each of the three target genes with the 3WJ-4 × Bro marker after incubation with DFHBI-1T, whereas leaves expressing marker-free genes or not incubated with DFHBI-1T showed no fluorescence (Fig. 5c and Supplementary Fig. 13), confirming that the 3WJ-4 × Bro/DFHBI-1T system could be used to report mRNA transcription in plant tissue in vivo. Furthermore, obvious cytoplasmic and nuclear localization signals were observed in leaf cells, demonstrating the distribution of three target mRNAs (Fig. 5c). Scanning of intracellular and extracellular fluorescence intensity (marked by blue lines in Fig. 5c) showed high S/B in the green fluorescence channel (Fig. 5d), which aided the accuracy of the 3WJ-4 × Bro/DFHBI-1T system.

To ensure accurate assessment of the expression levels of target mRNAs, we also detected the integrity of the 3WJ-4 × Bro-tagged

mRNAs through an in-gel imaging assay. We observed a single expected band on the gel for each 3WJ-4 × Bro-tagged mRNA (Supplementary Fig. 14). Sequencing results of the excised and eluted RNA bands revealed the full-length sequences of each 3WJ-4 × Bro-tagged RNA, indicating that the fluorescence came only from the 3WJ-4 × Bro-tagged mRNA. Thus, the 3WJ-4 × Bro/DFHBI-1T system accurately reported the expression and localization of the target mRNA in plant tissue.

Next, to further investigate the effect of the 3WJ-4 × Bro tag on the translation of the fusion mRNAs in plant, we visualized *mCherry-3WJ-4 × Bro* mRNA and its protein product in the leaves simultaneously through confocal microscopy. The bright green and red fluorescence signal were simultaneously observed in the *mCherry-3WJ-4 × Bro* channel and the mCherry channel, respectively, indicating that 3WJ-4 × Bro did not affect the function of mRNA as a genetically encoded template for protein translation in the plants (Fig. 5e). Moreover, we found no cells showing only green or red fluorescence alone, which indicated that 3WJ-4 × Bro was reliably linked to the mRNAs and vice versa. Thus, the spectroscopically and biochemically favorable characteristics of 3WJ-4 × Bro/DFHBI-1T, which include bright, stable fluorescence, high S/B contrast, and robust reporting of mRNA expression without perturbation of the target RNA's transcription, localization, or translation, qualify this system as an excellent protein-independent reporter system at the transcriptional level for plant genetic transformation.

To monitor the dynamic process of mRNA synthesis and transfer in plant cells, we transformed the construct expressing *AtCLE-3WJ-4 × Bro* into *N. benthamiana* leaf cells by *Agrobacterium* infiltration and then incubated leaf samples with 10-μM DFHBI-1T at 0, 24, 48, and 72-h post transformation, respectively. The imaging results showed no detectable fluorescence at 0-h post infiltration. *AtCLE-3WJ-4 × Bro* fluorescence

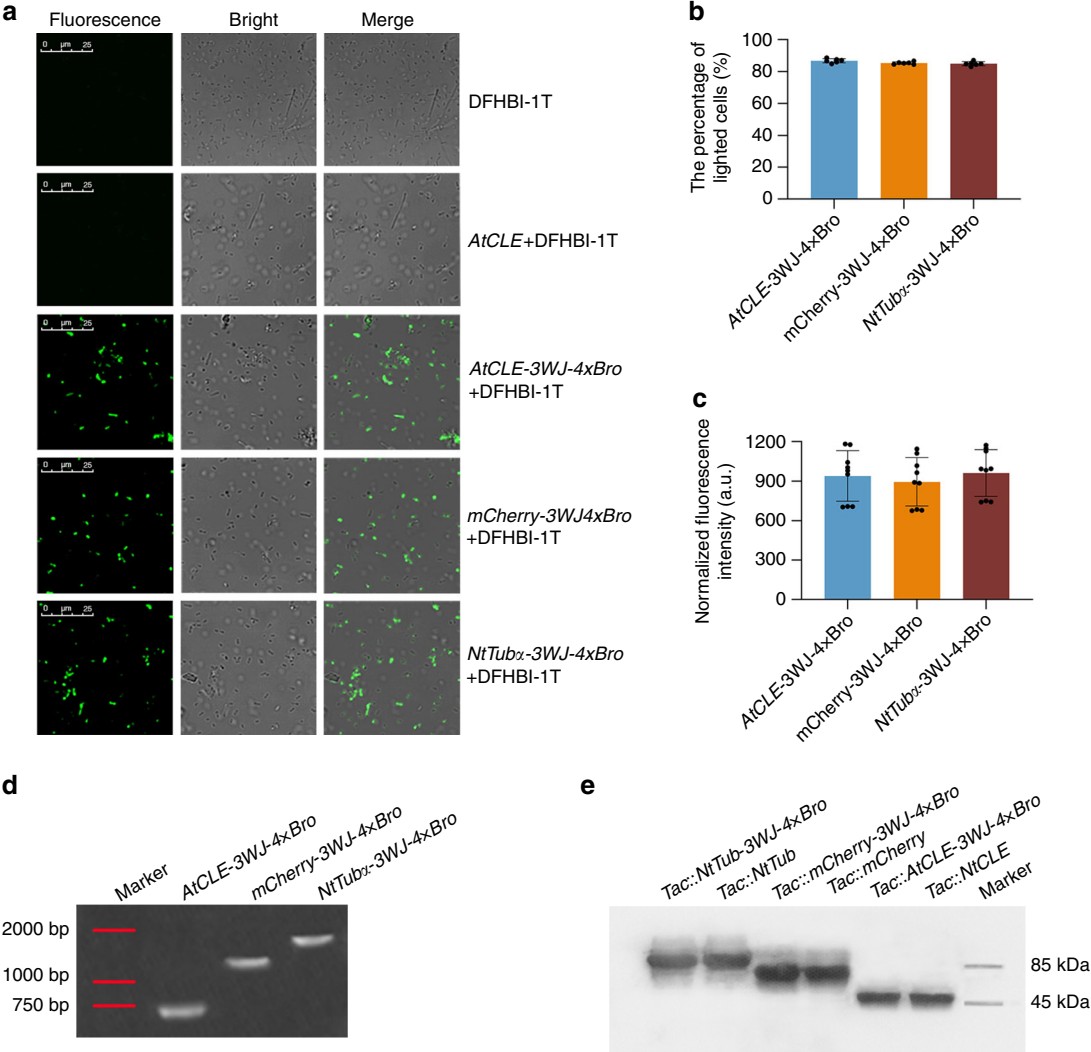

**Fig. 4 Performance of 3WJ-4 × Bro in *E. coli* cells for reporting the expression of various genes. a** Representative images of *E. coli* cells expressing three mRNAs tagged with 3WJ-4 × Bro after incubation with 10 μM DFHBI-1T. Scale bars, 25 μm. **b** The percentage of detectable fluorescing cells to the 500 total cells expressing each 3WJ-4 × Bro-tagged mRNAs. Error bars, mean ± s.d. (*n* = 6). **c** Fluorescence quantification of mRNAs tagged with 3WJ-4 × Bro in cells. Error bars, mean ± s.d. (*n* = 9). **d** Assessment of the integrity of three mRNAs with the 3WJ-4 × Bro tag in *E. coli* cells. Total cellular RNA was extracted and separated by urea denaturing gel electrophoresis. After washing, the gel image was stained with DFHBI-1T and photographed. Red lines indicate the locations of marker bands on the gel. **e** Immunoblot analysis of target proteins translated from 3WJ-4 × Bro-tagged mRNAs. Source data are provided as a Source Data Underlying **e**. All experiments were repeated three independent times with the same conclusion. Source Data are provided as a Source Data file.

was readily observed in the nucleus but not the cytosol at 24-h post infiltration, and bright green fluorescence gradually accumulated in the cytosol from 48- to 72-h post infiltration, reflecting a significant lag due to the time required for the *AtCLE* mRNA to be translocated from the nucleus to the cytoplasm (Fig. 5f). This dynamic was consistent with the known processes of mRNA synthesis and nucleus-to-cytosol transfer, showing that the 3WJ-4 × Bro/DFHBI-1T system can also be used for dynamic tracking of mRNAs.

**Detection of virus-mediated RNA delivery in plant using 3WJ-4 × Bro**. The virus-mediated RNA delivery system, which bypasses the requirement for producing transgenic plants, is widely used for gene overexpression and genome editing. TRV has been the most useful vector for this approach. However, the detection of RNA delivery still depends on RT-PCR, which is time consuming and inefficient. Consequently, we proposed a more efficient method for detecting RNA delivery using the 3WJ-

4 × Bro/DFHBI-1T system. Starting from a recombinant vector derived from the RNA2 genome of TRV, we created a vector to express *mCherry-3WJ-4 × Bro* mRNA under the control of the pea early browning virus (PEBV) promoter (Fig. 6a). Subsequently, the *Agrobacterium* cells respectively harboring the RNA1 genome and RNA2 vector were mixed and infiltrated into *N. benthamiana* leaves to reconstitute the TRV virus. We then detected fluorescence signals in both infiltrated and systemic leaves (non-infiltrated leaves of the same plant) under ultraviolet light by confocal microscopy at 5 days after infiltration.

After permeation of DFHBI-1T, infiltrated leaves showed visible green fluorescence upon exposure to ultraviolet light, but interestingly, weak fluorescence was also observed in systemic leaves (Fig. 6b), indicating that the *mCherry-3WJ-4 × Bro*-tagged mRNAs were delivered from the infiltrated leaves to systemic leaves by TRV. The fluorescence microscopy images further showed that *mCherry-3WJ-4 × Bro*-tagged mRNAs accumulated in the cytoplasm and nucleus of leaves of both types (Fig. 6b). The

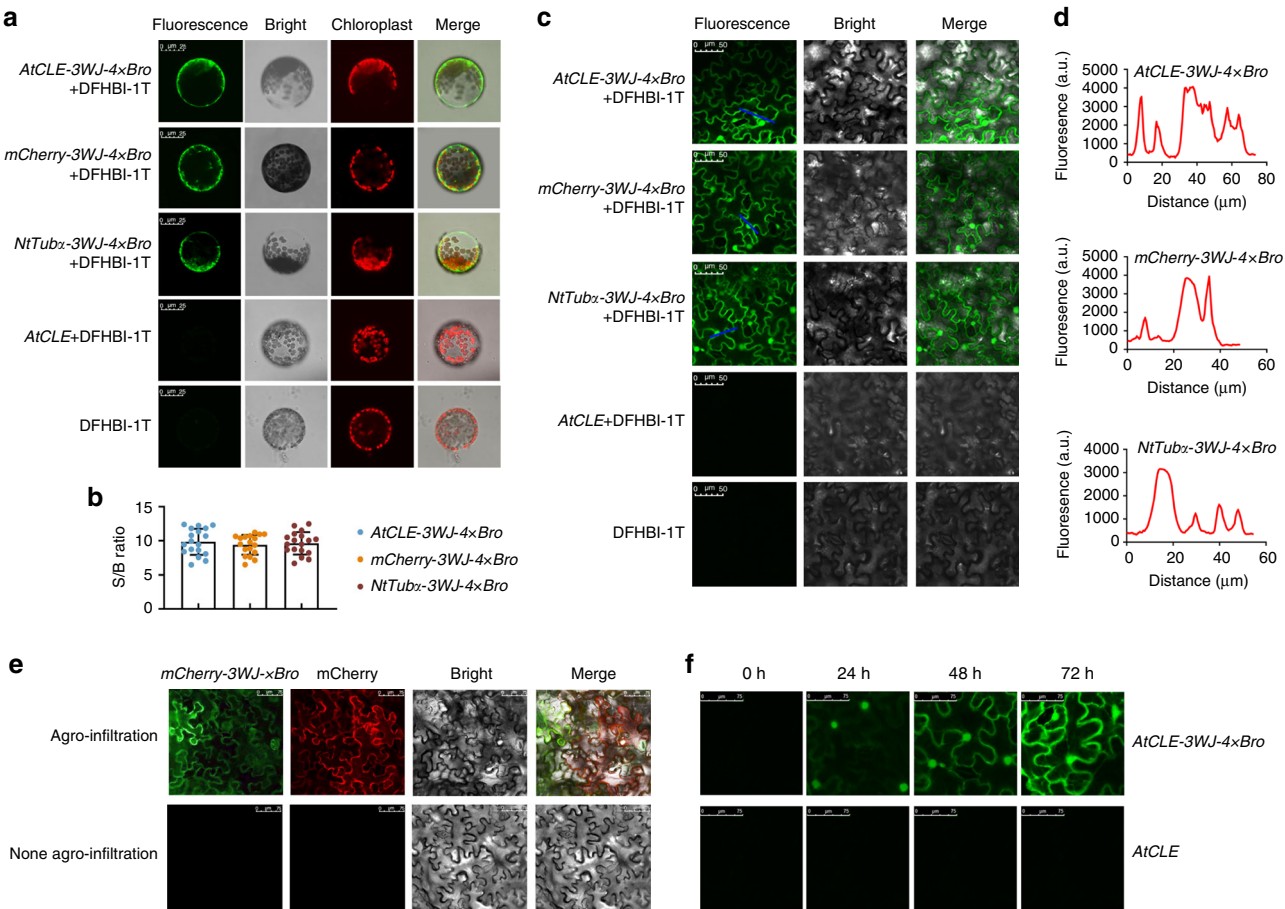

**Fig. 5 Use of 3WJ-4 × Bro to report target genes expression in plant. a** Confocal imaging of *N. benthamiana* protoplasts expressing three 3WJ-4 × Bro-tagged mRNAs. Scale bars, 25 μm. **b** Signal-to-background ratio of fluorescence in protoplasts incubated with 10 μM DFHBI-1T. Error bars, mean ± s.d. (*n* = 18). **c** Representative confocal images of *N. benthamiana* leaves expressing three 3WJ-4 × Bro-tagged mRNAs. Scale bars, 25 μm. **d** Detection of fluorescence and background in leaf cells. The blue lines in **c** were used for the line scan in **d**. **e** Simultaneous detection of *mCherry-3WJ-4 × Bro* mRNA and mCherry protein in *N. benthamiana* leaves by confocal laser microscopy. RNA of *mCherry-3WJ-4 × Bro* was imaged at 488-nm excitation and 527-nm emission and mCherry protein at 560-nm excitation and 610-nm emission. Scale bars, 75 μm. **f** Confocal microscopy detection of dynamic nuclear export process of *NtTubα* mRNA with 3WJ-4 × Bro in *N. benthamiana* leaf cells at 0, 24, 48, and 72-h post infiltration, respectively. Scale bars, 75 μm. All experiments were repeated three independent times with the same conclusion. Source Data are provided as a Source Data file.

red signal in the mCherry channel showed that mCherry protein was translated from *mCherry-3WJ-4 × Bro* and was mainly distributed in cytoplasm of both types of leaves (Fig. 6b). To further validate mRNA delivery, we performed RT-PCR amplification of *mCherry-3WJ-4 × Bro* from both inoculated and systemic leaves. A single expected band was detected in each leaf sample (Fig. 6c), and the sequencing data identified the same sequence of *mCherry-3WJ-4 × Bro* in two types of leaves.

**Performance of 3WJ-4 × Bro in plant stable transformation system.** Based on the outstanding performance of the 3WJ-4 × Bro/DFHBI-1T reporter system in plant transient transformation, we planned to use this reporter system for transgene identification and expression analysis of target genes in stable transformation. We selected *NtTubα* as a target gene and marked it with 3WJ-4 × Bro at its 3′-UTR. We then stably transformed the fusion gene *NtTubα-3WJ-4 × Bro* into *A. thaliana*. We screened 6000 transgenic $T_1$ seeds, which were germinated on MS culture medium containing 50 μg/mL kanamycin, resulting in 50 kanamycin-resistant $T_1$ transgenic seedlings. Leaves from kanamycin-resistant seedlings were excised for positive identification by genome PCR (gPCR) and fluorescence imaging after infiltration with 30-μM DFHBI-1T. gPCR analysis indicated that

45 of the 50 kanamycin-resisted seedlings were positive transgenic plants, and fluorescence imaging showed that 44 of these 50 seedlings had a bright green fluorescence (Fig. 7a, b). Moreover, all 44 fluorescent seedlings were gPCR-confirmed positive seedlings, but one of 45 gPCR-confirmed seedlings (designated N-F1) showed no fluorescence under the confocal microscope (Fig. 7b). To clarify this result, we assessed *NtTubα-3WJ-4 × Bro* expression in N-F1 seedlings by RT-PCR. No *NtTubα-3WJ-4 × Bro* expression was detected (Supplementary Fig. 15), indicating that *NtTubα-3WJ-4 × Bro* was successfully integrated into the genome but with its expression silenced. Thus, the 3WJ-4 × Bro/DFHBI-1T system can simultaneously and accurately report both the presence and expression of the target gene in transgenic plants.

To assess the genetic stability of 3WJ-4 × Bro in transgenic offspring, we grew six $T_2$ transgenic families derived from six $T_1$ plants in soil and subjected 766 $T_2$ seedlings to fluorescent identification. The results showed that 572 $T_2$ seedlings were fluorescent and 194 $T_2$ seedlings showed no fluorescence (Table 2). A $\chi^2$ test showed that the ratio of fluorescent plants to nonfluorescent plants fit the model of 3:1 Mendelian segregation in every $T_2$ family, indicating that all six $T_1$ transgenic plants harbor a single inserted copy of *NtTubα-3WJ-4 × Bro*

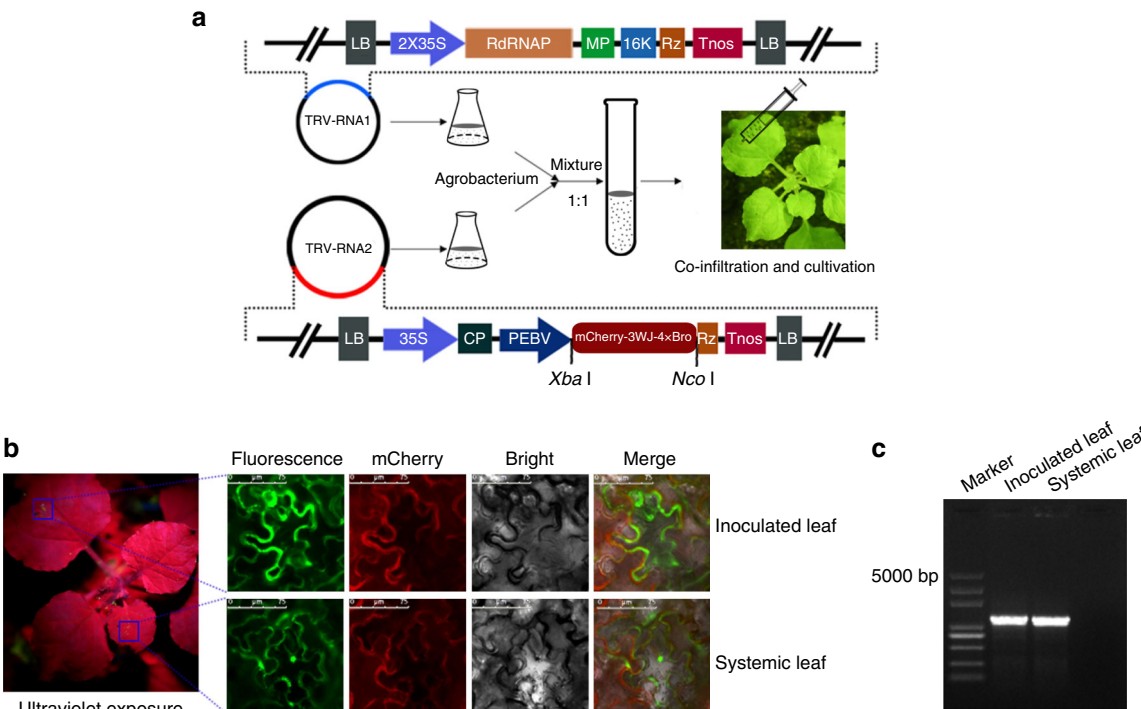

**Fig. 6 Detection of long-distance RNA delivery by 3WJ-4 × Bro in *N. benthamiana*. a** Schematic diagram of the *Agrobacterium* binary vector system containing the TRV RNA1 and RNA2 genomes with some modification, as well as the experimental scheme of the TRV-mediated target RNA delivery. RNA1 contains LB (left border), 2 × 35S (2 × CaMV *35S* promoter), RdRNAP (encoding RNA-dependent RNA polymerase), MP (encoding movement protein), 16k (encoding a cysteine-rich protein), Rz (self-cleaving ribozyme), Tnos (nopaline synthase terminator), and RB (right border). RNA2 contains LB, p35S, CP (encoding coat protein), PEBV (pea early browning virus promoter), Rz, Tnos, and RB. *mCherry-3WJ-4 × Bro* was cloned under the control of PEBV. The flask, tube, and syringe are drawn by software Photoshop CS6. **b** Fluorescence signals of *N. benthamiana* leaves under ultraviolet and confocal microscopes. *Agrobacterium* cultures carrying the engineered TRV RNA2 and RNA2 vectors were co-infiltrated into *N. benthamiana* leaves to express and transmit 3WJ-4 × Bro-tagged RNA (*mCherry-3WJ-4 × Bro*). At 5 days after *Agrobacterium* infection, both infiltrated and systemic leaves were sampled and then incubated with 10-μM DFHBI-1T at room temperature for 20 min. Scale bars, 75 μm. Three independent trials showed the same results. **c** Confirmation by RT-PCR of the presence of the fusion RNA (*mCherry-3WJ-4 × Bro*) in both infiltrated and systemic leaves. Source Data underlying **b** are provided as a Source Data file.

(Table 2). Furthermore, RT-PCR analysis of all the T$_2$ plants identified the 572 fluorescent plants as positive and the 194 nonfluorescent plants as negative (Table 2). These results indicated that 3WJ-4 × Bro linked with the target gene was stably heritable in the transgenic progeny, making 3WJ-4 × Bro an applicable gene marker for plant genetic engineering.

In addition, to investigate whether the 3WJ-4 × Bro/DFHBI-1T system could report target gene expression in different tissues of transgenic plants, we imaged the leaves, root elongation zone and root tip of transgenic *A. thaliana* seedlings by fluorescence microscopy. We detected green fluorescence signals in all three tissue types, which indicate that 3WJ-4 × Bro could report target gene expression in different plant tissues (Fig. 7c) and thus could potentially be used to conduct tissue-specific expression analysis of target genes. Furthermore, to compare the 3WJ-4 × Bro/DFHBI-1T reporter system and the GFP reporter system, we detected the fluorescence intensity of T$_1$ transgenic *A. thaliana* expressing *NtTubα-3WJ-4 × Bro* and *NtTubα-GFP*. Similar fluorescence intensities were observed between the two systems (Fig. 7d, e), suggesting that they possess similar sensitivity. These characteristics make 3WJ-4 × Bro/DFHBI-1T a compelling reporter system for transgene identification and gene expression analysis in plants.

## Discussion

Reporter systems are widely used in genetic engineering and promoter exploitation. The functions of conventional reporter systems depend on the expression of reporter genes such as *FP*, *GUS*, and *Luc*, which cause accumulation of foreign proteins in cells and hence may harm their growth and metabolism. Moreover, as has been reported before, these reporter systems have assorted limitations[4–6]. Consequently, we aimed to develop a protein-independent reporter system at the transcriptional level based on RNA fluorescence imaging that could be simultaneously used for identification of transgenic plants and expression analysis of target genes. To achieve this goal, it was critical to establish a precise technique for live-cell imaging of mRNA.

In our study, we set out to create fluorescence aptamers to cater to the need for RNA imaging in live plant cells and, further, to develop a reporter system for plant genetic engineering that acted at the transcriptional level. We chose to modify the F30 motif[29] rather than the tRNA motif to create a scaffold based on the consideration that the tRNA scaffold is susceptible to tRNA processing enzymes when an aptamer is inserted into its anticodon loop[30]. In addition, we optimized the Broccoli (Bro) sequence and selected as its fluorophore DFHBI-1T, a derivative of DFHBI, that has brighter fluorescence, non-toxicity, and higher permeability[27,31]. We designed a series of 3WJ scaffold aptamers for mRNA imaging in vitro and in vivo. Although 3WJ-Bro and F30-Bro have similar MFE secondary structures (Supplementary Fig. 1), 3WJ-Bro showed higher fluorescence intensity than F30-Bro (Fig. 1a, b), which we attributed to the fact that the modified 3WJ promoted Broccoli folding more effectively than F30 owing to differences in tertiary structure. 3WJ-4 × Bro showing brightest fluorescence intensity is mainly due to the increased number of

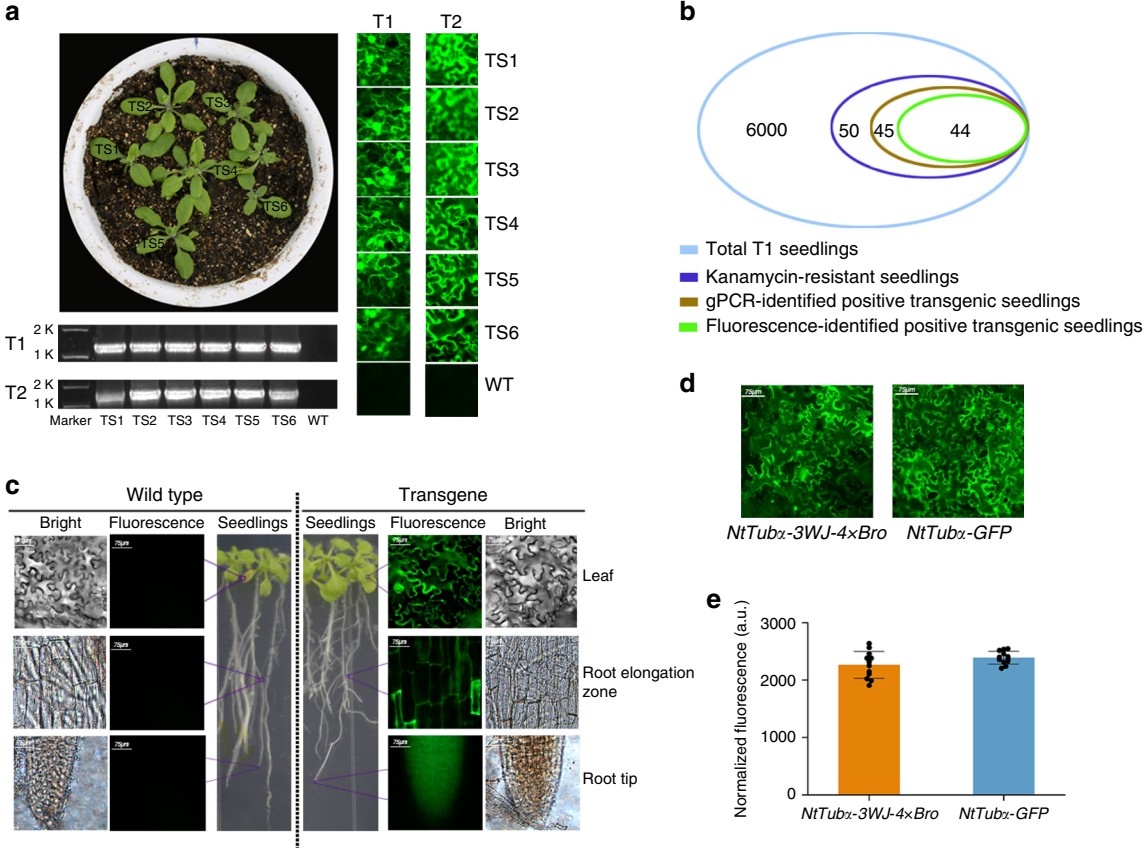

**Fig. 7 Identification of transgenic *A. thaliana* and expression analysis of target genes using the 3WJ-4 × Bro/DFBH-1T reporter system. a, b** Identification of T₁ and T₂ transgenic *A. thaliana*. **a** Representative image of transgene identification by 3WJ-4 × Bro fluorescence (right) and RT-PCR (bottom). TS transgenic lines, WT wild type, T₁, T₂ transgenic generations 1 and 2. **b** Numerical comparison of positive transgenic plants identified by the two methods above. **c** Expression of *NtTubα-3WJ-4 × Bro* in different tissues of transgenic *A. thaliana*. The wild type was used as control. **d** The imaging comparison of the 3WJ-4 × Bro/DFBH-1T system (left) and the GFP reporter system (right). Scale bars, 75 μm. **e** Quantitative comparison of fluorescence intensity between the 3WJ-4 × Bro/DFBH-1T and GFP reporter systems. Error bars depict the mean ± s.d. (*n* = 12). All experiments were repeated three independent times with the same conclusion. Source Data underlying **a–c, e** are provided as a Source Data file.

**Table 2 The identification of transgenic *Arabidopsis thaliana* T₂ plants by fluorescence and RT-PCR.**

| T₂ transgenic family | Total plant number | F-P plant number | F-N plant number | N-P plant number | N-N plant number | $\chi^2$ value for 3:1 |
|---|---|---|---|---|---|---|
| #1 | 122 | 93 | 0 | 0 | 29 | 0.098 |
| #2 | 114 | 86 | 0 | 0 | 28 | 0.012 |
| #3 | 133 | 100 | 0 | 0 | 33 | 0.003 |
| #4 | 139 | 105 | 0 | 0 | 34 | 0.022 |
| #5 | 116 | 86 | 0 | 0 | 30 | 0.046 |
| # 6 | 142 | 102 | 0 | 0 | 40 | 0.761 |
| Total | 766 | 572 | 0 | 0 | 194 | 0.044 |

F-P indicates both fluorescent and RT-PCR positive plants; N-N indicates neither fluorescent nor RT-PCR positive plants; F-N indicates fluorescent but RT-PCR negative plants; N-P indicates nonfluorescent but RT-PCR positive plants.
The one-sided $\chi^2$ test for 3:1 of F-P plants to N-N plants were conducted with significance level at 0.01. $\chi^2_{1,0.01} = 6.63$.

Broccoli units (four copies), while 3WJ-6 × Bro showed lower fluorescence compared to 3WJ-4 × Bro, which is possibly explained by that the 3WJ structure and linker 2 are not stable enough to effectively fold six copies of Broccoli. In addition, 3WJ-4 × Bro showing highest fluorescence stability is possibly attributed to its spatially stable structure enhanced by utilizing linker 1. To investigate the aptamers' ability to image mRNA in vivo, we fused each aptamer to the 3′-UTR rather than 5′-UTR of target genes to avoid the possibility that the presence of the aptamer tertiary structure fused to the 5′-UTR could abolish target gene

translation by hindering ribosome complex assembly[32,33]. Upon fusion with the mRNA *AtCLE*, we observed a significant decrease in fluorescence for F30-Bro, 3WJ-Bro, and 3WJ-6 × Bro, but not 3WJ-4 × Bro and 3WJ-2 × Bro, indicating that only 3WJ-4 × Bro and 3WJ-2 × Bro were stable enough to tag mRNA in vitro. In addition, in vivo assays revealed that although 3WJ-2 × Bro, 3WJ-4 × Bro, and 3WJ-6 × Bro all showed good potential for mRNA imaging in *E. coli*, only 3WJ-4 × Bro showed detectable fluorescence in *N. benthamiana*, indicating that stability of a given aptamer differed in different types of cells. This is probably the

main reason that, although aptamers such as Mango and Corn have been used for RNA imaging in *E. coli* and mammalian cells, there have been no reports of their application in plant cells.

To use 3WJ-4 × Bro as a marker for reporting gene expression, we investigated effects of different mRNAs on 3WJ-4 × Bro in vitro and in vivo. We established that 3WJ-4 × Bro could be used to image different mRNAs in both *E. coli* and plant cells, and the target proteins were detected in *E. coli* by immunoblotting and in plant cells by mCherry fluorescence (Figs. 4 and 5). Furthermore, no cleavage occurred between the mRNA and the tag in the 3WJ-4 × Bro-tagged mRNA series in either type of cell, indicating that 3WJ-4 × Bro fluorescence accurately reflects the expression and distribution of mRNA in live cells. By contrast, F29- and F30-scaffold Broccoli are reported to undergo differential cleavage in *E. coli* and mammalian cells[25], which implies that the structure of the 3WJ scaffold plays a pivotal role in maintaining the stability of both the Broccoli aptamer and RNAs of interest. Thus, 3WJ-4 × Bro is a favorable marker for reporting gene expression in plant cells without perturbation of the transcription or translation of target genes. Furthermore, our transient expression assays showed that the nuclear localization fluorescence signal of 3WJ-4 × Bro-tagged mRNAs was easily detected in *N. benthamiana* leaves, but not in *N. benthamiana* protoplasts. This difference was probably due to insufficient DFHBI-1T infiltration into protoplast nuclei during the relatively short incubation time required to avoid protoplast destruction. To assess the detection sensitivity of the 3WJ-4 × Bro/DFHBI-1T reporter system, we imaged the *NtTubα* mRNA tagged with 3WJ-4 × Bro and *NtTubα* protein with GFP under the same microscope and observed similar fluorescence intensities for both systems.

To assess the application of the 3WJ-4 × Bro/DFHBI-1T system, we used it to track an RNA virus invasion (Fig. 6), a process that poses severe hazards to the growth and development of plants, including crop plants. Although numerous molecular mechanisms of virus invasion have been described[34,35], direct imaging of RNA invasion dynamics is still lacking. In our study, we successfully detected the transfer of RNAs expressed by TRV from an inoculated leaf to other leaves, validating the utility of 3WJ-4 × Bro as a direct imaging tool for investigating RNA virus invasion and simultaneously facilitating research into the regulatory mechanism of long-distance transport of RNA. Homoplastically, the 3WJ-4 × Bro/DFHBI-1T system could also be employed to identify and monitor protein–RNA interactions[15] in real time in vivo. Moreover, the system is well suited for identifying transgenic plants (Fig. 7), for which it could be a competitive alternative to traditional diagnostic PCR. During the traditional transgenic generation screening process, gPCR identification is indispensable after antibiotic selection to ensure the insertion of the complete T-DNA into the plant chromosome as opposed to vector backbone integration, which may result from target gene escape[14]. In addition, qPCR analysis is required to evaluate silencing efficiency and/or transcript level in the transgenic plants. However, these molecular genetic tests are costly and time consuming[3]. Although GFP can be used as a marker for the rapid visual screening of transgenic plants[36,37], this approach requires constructing a fusion protein, which may negatively affect the biological function of target proteins. By contrast, 3WJ-4 × Bro provides a convenient means both to identify transgenic plants at the transcriptional level, without affecting target gene expression, and to estimate expression levels on the basis of fluorescence intensity as an alternative to qPCR. It could also be used to monitor the spatiotemporal-specific expression of a target gene during the course of a plant's lifetime.

The application of this system mainly depends on fluorescence intensity and stability. In our study, the obvious fluorescence was observed in plant cells, which is partly explained by the high accumulation transcripts of 3WJ-4 × Bro-tagged gene constitutively expressed under CaMV *35S*. However, endogenous gene showed a physiologically normal expression under the control of native promoter. Consequently, different types of endogenous promoter will be used to confirm potential utility of our system as a transcriptional reporter in plants in our further work. In addition, the fluorescence intensity is partly affected by the infiltration efficiency of DFHBI-1T into plant cell, which may be a limitation when comparing across different tissues or plants. This limitation could be got around if (i) plants cell membrane permeability could be changed by different solvents of DFHBI-1T solution; (ii) a derivative of DFHBI-1T, with high fluorescence and permeability, is developed.

In conclusion, we report protein-independent reporter system using a designed aptamer, 3WJ-4 × Bro, and its fluorophores, DFHBI-1T, for transgene identification and gene expression analysis in plants. The 3WJ-4 × Bro aptamer showed high folding efficiency and stability in plant cells, integrated into the plant genome with its target genes and co-expressed with them, and was stably inherited in future generations of transgenic lines through co-segregation with its target gene during the meiosis. Together, these properties make 3WJ-4 × Bro a desirable marker for plant genetic engineering and a useful tool for various types of functional genomics studies, including explorations of protein–RNA interaction, RNA–RNA interaction, RNA virus invasion, and promoter function.

## Methods

**Design and clone of aptamer-related sequences**. Based on the previously reported F30-Broccoli (F30-Bro), we modified the structure of the F30 scaffold to form a 3WJ scaffold and optimized the sequence of Broccoli (Supplementary Fig. 1). We designed a series of 3WJ-nBro aptamers by inserting Broccoli into two different specific sites of the 3WJ scaffold to construct 3WJ-Bro and 3WJ-2 × Broccoli (3WJ-2 × Bro) aptamers. To further improve the fluorescence intensity, we designed two specific linker sequences that provide two insert sites for 3WJ-2 × Bro to construct 3WJ-4 × Bro and 3WJ-6 × Bro (Supplementary Fig. 1). Secondary structure predictions of these RNA aptamers were calculated using the RNAfold web server application (http://rna.tbi.univie.ac.at/cgi-bin/RNAWebSuite/RNAfold.cgi). These aptamer DNA sequences were artificially synthesized (Supplementary Table 1) and individually cloned into the plasmid pBluescript II SK between *Sac* II and *Sac* I restriction site to form the constructs of pBluescript II SK-F30-Bro and pBluescript II SK-3WJ-nBro (Supplementary Fig. 16).

Total RNAs extracted from *A. thaliana* and *N. benthamiana* leaves by E.Z.N.A.® Total RNA Kit I (Omega, Cat. No. R6834) were respectively reverse transcribed to cDNA using TransScript® II One-Step gDNA Removal and cDNA Synthesis SuperMix (Transgen, Cat. No. AH311-02). The coding sequences of *AtCLE* (AT3G28455.1), *NtTubα* (LOC107826406), and *mCherry* were respectively amplified from *A. thaliana* cDNA, *N. benthamiana* cDNA, and plasmid pBI221-mCherry by specific primers (AtC F/AtC R, NtT F/NtT R, and mCh F/mCh R, respectively) (Supplementary Table 2), and then cloned into the plasmid pBluescript II SK between *Xba* I and *Sac* II restriction site to form the constructs of pBluescript II SK-AtCLE/mCherry/NtTubα (Supplementary Fig. 17).

The recombinant pBluescript II SK-AtCLE-3WJ-nBro (Supplementary Fig. 18) and pBluescript II SK-AtCLE/mCherry/NtTubα-3WJ-4 × Bro (Supplementary Fig. 19) were constructed by enzyme digestion and ligation (*Xba* I and *Sac* I).

**In vitro transcription and fluorescence measurement**. F30-Bro, 3WJ-nBro, AtCLE-3WJ-nBro, and AtCLE/mCherry/NtTubα-3WJ-4 × Bro were amplified from their respective recombinant pBluescript II SK plasmids by PCR using the universal primers of M13 F and M13 R (Supplementary Table 2). The RNAs were transcribed in vitro using T7 RNA polymerase at 37 °C for 16 h in 40 mM Tris-HCl, 30 mM MgCl₂, 2 mM spermidine, 1 mM dithiothreitol (DTT), 5 mM rNTPs, 1 U/μL inorganic pyrophosphatase and 4 U/μL T7 RNA polymerase (pH 7.9). The transcripts were purified on a denaturing 6% polyacrylamide gel (acrylamide: bisacrylamide = 29:1) containing 6 M urea in TBE (Tris, borate, ethylenediaminetetraacetic acid (EDTA)) buffer[38]. The RNA was excised and eluted in RNA elution buffer (40 mM Tris-HCl, pH 8.0, 0.5 M sodium acetate, 0.1 mM EDTA) and precipitated with ethanol[39]. All transcripts were dissolved in DEPC-purified water, and the purity and concentration of RNAs were measured with a NanoDrop ND-2000 spectrophotometer. Subsequently, 1 μg transcribed RNA was incubated with 10 μM of fluorophore DFHBI-1T ((*Z*)-4-(3,5-difluoro-4-hydroxybenzylidene)-2-methyl-1-(2,2,2- trifluoroethyl)-1*H*-imidazol-5(4*H*)-one) in 50 μL 1 × HEPES buffer (PH 7.4) in a centrifuge tube at 25 °C for 20 min. DFHBI-1T was purchased

from LUCERNA[40].The excitation spectra of the complexes were scanned from 400- to 540-nm wavelengths at 527-nm emission and the emission spectra of that were scanned from 480 to 650 nm at 460-nm excitation using a fluorescence spectrophotometer (Hitachi F-7000). The extinction coefficient was calculated based on the method described by Song et al.[41]. The fluorescence of each aptamer–DFHBI-1T complex was imaged with a Blue Light Gel Imager[19]. The fluorescence intensity was measured at 488-nm excitation and 527-nm emission wavelengths.

**Live-cell imaging mRNA in *E. coli*.** Recombinant pBluescript II SK(+) plasmids containing F30-Bro, 3WJ-nBro, AtCLE-3WJ-nBro, and AtCLE/mCherry/NtTubα-3WJ-4 × Bro were transformed into *E. coli* strain BL21 (DE3) cells and then grown overnight on Luria Broth solid medium containing 50 µg/mL ampicillin. Single positive clones identified by PCR were inoculated into Luria Broth liquid medium containing 50 µg/mL ampicillin and grown to exponential phase 37 °C at 200 rpm in a biochemical incubator, the cultures were then supplemented with 1 mM iso-propyl β-D-1-thiogalactopyranoside (IPTG) and subjected to shaking incubation at 16 °C at 160 rpm for the inductive expression of mRNAs with 3WJ-4 × Bro tag. The cells were collected from 100 µL cultures and washed once with 1 mL M9 medium (47.75 mM $Na_2HPO_4$, 22.04 mM $KH_2PO_4$, 8.56 mM NaCl, 18.7 mM $NH_4Cl$, 2 mM $MgSO_4$, 0.1 mM $CaCl_2$). The washed cells were resuspended in M9 medium to an $OD_{600}$ of 2 and incubated with 10 µM DFHBI-1T for 30 min at 25 °C. Aliquots of 10 µL were transferred to a 384-well microtiter plate, and fluorescence intensity at 25 °C was measured at 488-nm excitation and 527-nm emission wavelengths. Then, 5 µL of the above cells were diluted to 20 µL with M9 medium. Aliquots of 10 µL were transferred to a glass slider and covered with a coverslip. The images of *E. coli* were taken at room temperature using a confocal microscope (objective ×63, N.A. 1.46, excitation 488 nm, emission 527 nm) and analyzed with LAS AF Lit software[42].

**Prokaryotic expression and immunoblotting.** *AtCLE-3WJ-4 × Bro*, *mCherry-3WJ-4 × Bro*, and *NtTubα-3WJ-4 × Bro* were respectively amplified from recombinant plasmids pBluescript II SK-AtCLE/mCherry/NtTubα-3WJ-4 × Bro by specific primers of pM-At-3WJ F/pM-At-3WJ R, pM-mC-3WJ F/pM-mC-3WJ R, and pM-Nt-3WJ F/pM-Nt-3WJ R (Supplementary Table 2) and subcloned into prokaryotic expression pMol-c2x plasmids between *Sac* I and *Xba* I site (Supplementary Fig. 20). The recombinant pMol-c2x were transformed individually into *E. coli* strain Rosetta (DE3). Single positive clones identified by PCR were inoculated into Luria Broth liquid medium supplemented with 50 µg/mL ampicillin and grown overnight at 37 °C at 200 rpm, and then transferred into fresh Luria Broth medium containing 50 µg/mL ampicillin at a ratio of 1:100 in a shaking incubator at 30 °C at 180 rpm. IPTG was added to the culture medium at 1 mM final concentration to induce protein expression and the cultures were further incubated for 10 h at 16 °C at 160 rpm, when the $OD_{600}$ reached 0.6–0.8. The cells were collected by centrifugation at $6200 \times g$ for 2 min at 4 °C, resuspended in the buffer B1 (20 mM Tris-HCl, 500 mM NaCl, 1 mM EDTA, pH 8.5), and subjected to ultrasonic disruption. The cell lysate was then separated by centrifugation at $9600 \times g$ for 10 min at 4 °C. The precipitate was washed twice with buffer B1 and resuspended in the same volume of buffer B3 (20 mM Tris-HCl, 500 mM NaCl, 1 mM EDTA, 40 mM DTT, pH 8.5). The protein expression level and purity were estimated using SDS-PAGE using a 10% tricine SDS-PAGE gel and Coomassie blue staining[43].

For immunoblot analysis, the induced cells were collected and washed with ice-cold 1 × phosphate-buffered saline. The cells were lysed with ice-cold lysis buffer (50 mM Tris-HCl (pH 7.5), 150 mM NaCl, 1 mM EDTA, 1% NP-40, 2 × protease inhibitor cocktail (Invitrogen)) for 30 min on ice. The lysate was selected by centrifugation at $11,600 \times g$ for 10 min at 4 °C. Protein lysate samples were separated using 10% SDS-PAGE and then transferred onto polyvinylidene difluoride membranes (EMd Millipore, Billerica, MA, USA). The membranes were blocked using 5% skim milk for 2 h at room temperature. After washing with 1 × TBST buffer (50 mM Tris base, 155 mM NaCl, pH 7.6, 0.05% Tween 20), the membranes were incubated with anti-MBP mouse monoclonal antibody (Cat. No. HT701-01, Transgen, China) at 1:1000 dilution overnight at 4 °C. The membranes were washed with 1 × TBST buffer, followed by goat anti-mouse IgG (H + L)-HRP (Cat. No. HS201-01, Transgen, China) at dilution of 1:4000 incubation for 1 h at room temperature.

**Live-cell imaging 3WJ-4 × Bro-tagged mRNA in plant protoplasts.** *AtCLE-3WJ-4 × Bro, mCherry-3WJ-4 × Bro,* and *NtTubα-3WJ-4 × Bro* were amplified from their respective recombinant pBluescript II SK plasmids by specific primer pairs pB-At F/pB-At R, pB-mC F/pB-mC R, and pB-Nt F/pB-Nt R (Supplementary Table 2) and subcloned into pBI221 plasmids between *Xba* I and *Sac* I site (Supplementary Fig. 21) for protoplast transformation. The *N. benthamiana* protoplasts were obtained by the protocol described by Yoo et al.[44]. Then, 100 µL protoplast solution ($1 \times 10^4$ protoplasts), 10 µL plasmid DNA (1 µg/µL), and 110 µl PEG solution (PEG4000, 0.2 M mannitol, 100 mM $CaCl_2$) were combined in 2-mL round-bottomed microcentrifuge tubes and mixed gently. The transfection process was stopped by adding 440 µL W5 solution (2 mM MES (pH 5.7), 154 mM NaCl, 125 mM $CaCl_2$, and 5 mM KCl) to transfection mixture after 15 min incubation at

room temperature. The mixture was centrifuged at $100 \times g$ for 3 min at room temperature to remove the supernatant. The protoplasts were resuspended in 1 mL WI solution (4 mM MES (pH 5.7), 0.5 M mannitol, 20 mM KCl) and incubated at room temperature for 12 h. The protoplasts were centrifuged at $100 \times g$ for 2 min to remove the supernatant and 200 µL WI solution was added to resuspend the protoplasts. DFHBI-1T at a final concentration of 10 µM was added into the protoplast solution for additional 30-min incubation. Live cellular images of mRNAs tagged with 3WJ-nBro were viewed with a confocal microscope (objective ×63, N.A. 1.46, excitation 488 nm, emission 527 nm) and analyzed with LAS AF Lit software. The fluorescence intensity was measured with a fluorescence plate reader: 100 µL of the treated protoplast solution above was transferred to a 384-well plate, and fluorescence intensity at 25 °C was measured at 488-nm excitation and 527-nm emission wavelengths.

**Expression of 3WJ-4 × Bro-tagged mRNA in *N. benthamiana* leaves.** Recombinant pBI121 binary plasmids containing *AtCLE-3WJ-4 × Bro*, *mCherry-3WJ-4 × Bro*, and *NtTubα-3WJ-4 × Bro* were constructed (Supplementary Fig. 22) and transformed into *Agrobacterium tumefaciens* strain GV3101. Positive clones were picked from YEP agar plates containing 50 µM kanamycin and 35 µM rifampicin and identified by PCR, and were then inoculated into 15 mL YEP medium containing the same antibiotics for incubation at 28 °C overnight at 180 rpm in a shaker. The cells were pelleted by centrifugation at $2500 \times g$ for 10 min at room temperature when $OD_{600}$ reached 0.8 and resuspended in agromix buffer (10 mM MES (pH 5.8), 10 mM $MgCl_2$, 200 µM acetosyringone) to an $OD_{600}$ of 1.0. The mixture was infiltrated into the leaves of *N. benthamiana* plants using a syringe without a needle and the infiltrated area of the leaves was marked[45]. After 3 days of cultivation at 25 °C in a growth chamber with 16 h/8 h (light/dark), the leaves were incubated with 10 µM DFHBI-1T by infiltration for 16 h. The mRNAs tagged with 3WJ-4 × Bro were detected on the basis of their fluorescent signals using a confocal microscope (objective HCX PL APO CS 20.0 × 0.70 DRY UV, excitation 488 nm, emission 527 nm).

**Detection of virus-mediated RNA delivery by the reporter system.** The sequence of *mCherry-3WJ-4 × Bro* was amplified from recombinant plasmids pBluescript II SK-*mCherry*-3WJ-4 × Bro by specific primers TRV2-mC F and TRV2-mC R (Supplementary Table 2) and subcloned into pTRV2 plasmid between *Xba* I and *Nco* I site (Fig. 6a). The pTRV1 and recombinant pTRV2 were individually transformed into *Agrobacterium* GV3101. The positive *Agrobacterium* cells were pelleted from the medium by centrifugation and resuspended in infiltration buffer (10 mM $MgCl_2$, 10 mM MES, 200 µM acetosyringone) to $OD_{600}$ 2.0. *Agrobacterium* cells harboring pTRV1 was mixed with that harboring recombinant TRV2 or nonrecombinant TRV2 (as control) at a 1:1 ratio, and each mixture was infiltrated into the leaves of *N. benthamiana* plants with a syringe without needles after 2 h of incubation at room temperature in darkness. The plants were grown in a growth room with a 16 h/8 h (light/dark) photoperiod at a light intensity of 10,000 lux at 24 °C for 5 days[46]. The leaves, including new leaves, were cut off and infiltrated with 10 µM DFHBI-1T for 16 h and then the distribution and expression of mRNA were detected by fluorescence using a confocal microscope (objective HCX PL APO CS 20.0 × 0.70 DRY UV, excitation 488 nm, emission 527 nm). Total RNA was extracted from root, shoot, and leaf samples. The relative expression was determined by qRT-PCR.

**Identification of stably transformed *A. thaliana* by the reporter system.** *Agrobacterium* GV3101 containing recombinant binary plasmids pBI121-*NtTubα*-3WJ-4 × Bro were grown to $OD_{600}$ of 1.6 at 28 °C at 180 rpm in a shaker. The cells were pelleted by centrifugation and resuspended in infiltration medium (5% sucrose, 0.02% Silwet L-77) to a final $OD_{600}$ of 1.0 and then incubated at room temperature in darkness for 2 h. *A. thaliana* (Columbia ecotype) inflorescences were immersed in *Agrobacterium* solution for 10 s twice and covered with plastic bags to retain humidity[47]. After 16 h of cultivation in darkness, the plants were transferred to a light chamber (with a 16-h light/8-h dark photoperiod) for further growth. The seeds of $T_0$ generation were obtained screened on the solid MS medium containing 30 µg/mL kanamycin. Two-week-old kanamycin-resistant seedlings were respectively transferred to soil and MS medium for 10 days of growth. The tissues (leaf, root elongation zone, and root tip) excised for a gPCR identification using the primer pairs 35S/NOS (Supplementary Table 2) and a fluorescence detection using a confocal microscope (objective HCX PL APO CS 20.0 × 0.70 DRY UV, excitation 488 nm, emission 527 nm) after infiltration with 30 µM DFHBI-1T. Besides, the total RNA was extracted form leaves of fluorescently transgenic plant and the RT-PCR were conducted by primers RT F/RT R (Supplementary Table 2) for reconfirming the expression of *NtTubα-3WJ-4 × Bro* in these transgenic plants.

**Integrity analysis of mRNAs with 3WJ-4 × Bro tag.** The integrity of the transcripts of 3WJ-4 × Bro-tagged genes expressed in live cells was detected by in-gel imaging of fluorescence RNAs. Total RNA was extract from live cells (*E. coli*, *A. thaliana*, *N. benthamiana*) and isolated on an 8% urea gel in TBE buffer. The gel was washed three times with RNase-free water for 15 min to remove the urea, and stained for 20 min at room temperature in 10 µM DFHBI-1T in 1 × HEPES (pH

7.4). The gel was imaged on a ChemiDoc MP (Bio-Rad) with a 488-nm excitation and 527-nm emission[2]. The expected band was excised from the gel and the eluted mRNA was reverse transcribed to cDNA using a High Capacity cDNA Reverse Transcription Kit (ThermoFisher). The cDNA was then sequenced to confirm the integrity of the mRNA with 3WJ-4 × Bro tag.

**Reporting summary**. Further information on research design is available in the Nature Research Reporting Summary linked to this article.

## Data availability

Data supporting the findings of this work are available within the paper and its Supplementary Information files. A reporting summary for this Article is available as a Supplementary Information file. The datasets generated and analyzed during the current study are available from the corresponding author upon request. The data that support the findings of this study are available from the corresponding author upon reasonable request. The data underlying Figs. 1, 2b–e, 3–5, 6b, 7a–c, and 7e, as well as Supplementary Figs. 4, 5, 10, 12, and 14 are provided as a Source Data file. Source data are provided with this paper.

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

## Acknowledgements

This work was supported by the Science and Technology Department of Sichuan Province (Grants 2017SZ0181 and 2018NZDZX0003) and the National Key R & D Program (Grant 2018YFC1802605), People's Republic of China. This research was also supported by the Fundamental Research Funds for the Central Universities (SCU2019D013).

## Author contributions

Conceptualization, Y.Z. and C.F.; funding acquisition, Y.Z. and R.W.; investigation, J.B., Y.L. and X.W.; data analysis, J.B., Y.L., X.W., S.L., M.Y., M.L. and Y.Z.; visualization, G.L., J.Y., H.Y. and M.Z.; writing original draft, J.B. and R.W.; reviewing and editing, Y.Z., C.F., W.W. and M.W.

## Competing interests

The authors declare no competing interests.
