## [Peer Review File · Nature Communications]

Reviewers' Comments:

Reviewer #1:

Remarks to the Author:

Here are my comments on the paper entitled 'A novel protein-independent reporter system based on the RNA fluorescence aptamer for plant genetic engineering'.

In this report, the author created a protein-independent reporter system based on the RNA fluorescence aptamer using the RNA three-way junction to report the expression of target genes both in *E. coli* and leaf cells. The new 3WJ reporter system is reported to be able to reveal the transgenic lines identification and expression pattern of the target gene at transcriptional level. In combination with virus-mediated gene expression system, the 3WJ reporting system was used to directly visualize the delivery of target RNA in plant. The application of this system using RNA as indicator is interesting since the field of RNA is more and more popular and the application in plant is relatively new. But the manuscript needs significant revisio, as indicated below before reconsideration for publication.

Title: "Novel" should be removed from the title, then "A protein-independent reporter based on RNA 3-way junction (or 3WJ) and fluorescence aptamer for plant genetic engineering".

Abstract: a more concise abstract is requested.

1. A 2D structure including the aptamer and the 3Wj sequence should be provided in Fig 1.
2. Acronyms should be defined before use (i.e., line 680 the 'SEM').
3. This is a method paper, more quantitative data should be provided. The authors modified the previously reported 3WJ-Broccoli and mentioned that their structure was better. They indicated that their modified structure can achieve two-fold higher fluorescence than the reported one. Since two-fold is not a high enhancement, it is very hard to interpret if these values are significant. If they want to use for comparison, a consequence of variation between measurements should be reported with statistical analysis.
4. Some results are stated without statistical analysis and it's so vague. For example, line 141, 'Besides, the rate of fluorescence dissipation of 3WJ-BRO tagged mRNAs decreased with increasing molecular weight of mRNA, indicating that the mRNA with larger molecular weight may contribute to the stability of 3WJ-BRO (Fig. 1e). How did you calculate and what is the magnitude and rate of fluorescence? How many trials have you done? Are they statistically different?
5. Some data indicated the number of trials, but others are not. For example, line 680, the data expressed as mean \pm SEM for six independent experiments, whereas for the data in figure 1e, it is not clear how many trials you have done.
5. Line 147, it is mentioned that the 3WJ-BRO tagged mRNAs could be easily imaged and detected at the level of 9.375 nmol, indicating a higher sensitivity. Please provide more information on how to reach the conclusion of increasing

sensitivity.

6. A paper reported the similar method should be cited and discussed: Reif R et al. Fluorogenic RNA Nanoparticles for Monitoring RNA Folding and Degradation in Real Time in Living Cells. *Nucleic Acid Ther.* 2012 22(6):428-37.

Reviewer #2:

Remarks to the Author:

The ms contains some potentially interesting information that could be helpful to plant biologists. The problems I have with the ms/study in its present form is multi-tiered. 1. The writing is suboptimal. The English is non-standard. I really had a difficult time reading it. 2. It doesn't appear that the authors are plant biologists or use the same words to describe plant biology as plant biologists do. For example, the authors start the ms with some very tired and rejected (e.g., lines 46-47) and with no reference to support the statements. Moreover, the topic of the paper is not going to replace selectable and scorable DNA markers in plant biotech. 3. The work is largely derivative. The fluorescent aptamer is not really novel in concept and form. Their system is based off the broccoli system and in some places they have not given the correct citation of preceding work; e.g., line 109. 4. The experiments and results are insufficient to justify the conclusions. I don't think we're ever told about the spectral characteristics of the system; e.g., optimal excitation, extinction coefficient, etc. The experiments mainly rely on transient expression in *benthi*, which is ok, but plant biologists will likely be most interested in the system for analysis in stably transgenic plants. Here, the authors recovered transgenic *arabidopsis* plants, but really very little data are shown and the data shown aren't very thorough or convincing that their system is transformative. Therefore, based on these issues, I am not enthusiastic about this ms.

Reviewer #3:

Remarks to the Author:

This manuscript describes a mRNA detecting marker system for plants. The authors designed a RNA fluorescence aptamer (3WJ-BRO) based on broccoli RNA aptamer. They have shown that the tetrameric version is more sensitive than broccoli aptamer (which is more sensitive than spinach aptamer) for mRNA detection and tracking. Spinach and Broccoli aptamers have been used successfully for imaging of RNA in live bacteria and for tracking the transcription products of RNA pol III in mammalian cells. This type of labeling is not sufficiently sensitive to enable the detection of lower abundance mRNAs. To increase the sensitivity, people have shown to arrange multiple Spinach or Broccoli aptamers in tandem array to

increase fluorescence. The authors tagged the 3WJ-BRO on the 3' end of three genes and evaluated the fluorescence in vitro, in DNA-transfected *N. benthamiana* plant protoplasts, in Agro-infiltrated *N. benthamiana* plant leaves, and in transgenic *Arabidopsis* plants. This application with enhanced broccoli RNA-fluorophore pair has not been reported in plant system. RNA reporter system is a very important tool in functional gene study. The use of fluorescent reporter for transcription study in live cells, if sensitive enough, can really positively impact molecular biological research. However, this reviewer is not completely convinced that the fluorescence images reported in this manuscript were the true mRNA localization or simply artifacts of the experimentation.

- Figure 1d, the first and third samples have same label. I think the first sample should be "3WJ-BRO" alone. Please check and correct.
- Figure 3, Why are all green fluorescence images showing the localization on cell membranes? Please include images with and without DFHBI-1T for all constructs. What is the size bar length?
- Figure 3, adding DFHBI-1T seems to influence the chloroplast distribution in protoplast, comparing all DFHBI-1T samples with the one sample without DFHBI-1T. It seems that in DFHBI-1T (+) samples, the chloroplasts are moving to the cell membranes.
- Figure 3, please show the mCherry expressing protoplasts, and if it overlay image with mCherry-3WJ-BRO.
- Figure 4b is redundant with Figure 2c. The marker lane is empty.
- Figure 4e, need images with plant not Agro-infiltrated.
- Figure 5, need to see mCherry expression images in these lines. All confocal images need to be enlarged to allow readers to view the details.
- Figure 6a, the confocal images should be enlarged to allow readers to view the cellular details. There is no need to show the *Arabidopsis* plants and their leaf sampling information.
- Figure 6b, need to include images from non-transgenic cells.
- The manuscript needs to be substantially improved for English and grammar. Sometimes the presentation is so poor, it was very hard to even understand what the authors try to say.

Point-by-point response letter

Dear reviewers

We'd like to thank you very much for your constructive comments on our manuscript, and have performed some experiments to address the questions you raised. Therefore, we carefully revised the manuscript and provide this point-by-point letter as follows.

Point-by-point response to the reviewers

Reviewer #1 (Remarks to the Author):

Here are my comments on the paper entitled 'A novel protein-independent reporter system based on the RNA fluorescence aptamer for plant genetic engineering'.

In this report, the author created a protein-independent reporter system based on the RNA fluorescence aptamer using the RNA three-way junction to report the expression of target genes both in E. coli and leaf cells. The new 3WJ reporter system is reported to be able to reveal the transgenic lines identification and expression pattern of the target gene at transcriptional level. In combination with virus-mediated gene expression system, the 3WJ reporting system was used to directly visualize the delivery of target RNA in plant. The application of this system using RNA as indicator is interesting since the field of RNA is more and more popular and the application in plant is relatively new. But the manuscript needs significant revision, as indicated below before reconsideration for publication.

Thank you very much for your positive assessment and for giving us the opportunity to clarify these points.

Title: "Novel" should be removed from the title, then "A protein-independent reporter based on RNA 3-way junction (or 3WJ) and fluorescence aptamer for plant genetic engineering".

Abstract: a more concise abstract is requested.

Following your suggestion, we have changed the title of our manuscript as "A protein-independent reporter system based on the RNA fluorescence aptamer for plant genetic engineering". Also we revised the abstract and make it more concise and accurate.

1. A 2D structure including the aptamer and the 3Wj sequence should be provided in Fig 1.

We have changed layout of Fig. 1 and provided the new data including the *in vitro* comparison of new designed RNA aptamers. And, we have presented 2D structure including the aptamer and the 3Wj scaffold in Fig. S1 and Fig. S2.

2. Acronyms should be defined before use (i.e., line 680 the 'SEM').

Thanks for your suggestions. All the acronyms have been defined at their first appearance in our revised manuscript.

3. This is a method paper; more quantitative data should be provided. The authors modified the previously reported 3WJ-Broccoli and mentioned that their structure was better. They indicated that their modified structure can achieve two-fold higher fluorescence than the reported one. Since two-fold is not a high

enhancement, it is very hard to interpret if these values are significant. If they want to use for comparison, a consequence of variation between measurements should be reported with statistical analysis.

We performed fluorescence image and quantification by six independent trials and calculated the mean of fluorescence intensity. Moreover, we did the statistical analysis by one-way ANOVA and the data are shown in Fig. 1b in revised manuscript.

4. Some results are stated without statistical analysis and it's so vague. For example, line 141, 'Besides, the rate of fluorescence dissipation of 3WJ-BRO tagged mRNAs decreased with increasing molecular weight of mRNA, indicating that the mRNA with larger molecular weight may contribute to the stability of 3WJ-BRO (Fig. 1e). How did you calculate and what is the magnitude and rate of fluorescence? How many trials have you done? Are they statistically different?

Sorry for the confusion. We have provided statistical analysis for all applicable experiments in the revised manuscript. We have changed previous Fig. 1e to Fig. 3d in new version. In this figure, we try to estimate the effect of different target mRNA on fluorescence stability of 3WJ-4×Bro (namely previous 3WJ-BRO). The fluorescence intensity of 3WJ-4×Bro tagged RNAs were measured by a Fluorescence Spectrophotometer (Hitachi F-7000) at an excitation wavelength of 488 nm every hour post-administration. The relative rate of fluorescence decay was calculated after normalization of remained to its respective initial fluorescence intensity. Six trails we have conducted and the statistical difference was found regarding fluorescence decay at 5h after excitation. Result showed an increased fluorescence stability for 3WJ-4×Bro with the increasing molecular weight of mRNAs over time, indicating that mRNA with larger molecular weight might contribute to the stability of 3WJ-4×Bro (Fig. 3d). These statements located in line 176-189, page 6 in revised version.

5. Some data indicated the number of trials, but others are not. For example, line 680, the data expressed as mean ± SEM for six independents experiments, whereas for the data in figure 1e, it is not clear how many trials you have done.

All the data presented were derived from 6 trials and the sample information has been supplemented at the end of each figure legend in the revised version.

5. Line 147, it is mentioned that the 3WJ-BRO tagged mRNAs could be easily imaged and detected at the level of 9.375 nmol, indicating a higher sensitivity. Please provide more information on how to reach the conclusion of increasing sensitivity.

To compare the detection limit of the new 3WJ-4×Bro (previous 3WJ-BRO) and original F30-Bro (previous F30-broccoli), we performed a serial dilution of in vitro transcribed 3WJ-4×Bro and F30-Bro tagged RNAs, the detection limit of 3WJ-4×Bro tagged RNA is 9.375 nmol, while the previous report showed 32 nmol detection limit for F30-Bro tagged RNAs, which indicated that 3WJ-4×Bro showed a higher sensitive to DFHBI-1T compared to F30-Bro. The detailed information has been provided in Fig. S7 and Fig. 3a in the revised manuscript.

6. A paper reported the similar method should be cited and discussed: Reif R et al. Fluorogenic RNA Nanoparticles for Monitoring RNA Folding and Degradation in Real Time in Living Cells. Nucleic Acid Ther. 2012 22(6):428-37.

Sorry for the negligence. We added this reference shown as Ref [25] in the revised manuscript.

Reviewer #2 (Remarks to the Author):

The ms contains some potentially interesting information that could be helpful to plant biologists.

Thanks for your encouraging comment.

The problems I have with the ms/study in its present form is multi-tiered. 1. The writing is suboptimal. The English is non-standard. I really had a difficult time reading it.

We deeply apologize for the suboptimal presentation. In the revised manuscript, we tried our best to correct the ineffective and inaccurate usage, and also make it more concise. In addition, we invited a native speaker to improve the English of the paper.

2. It doesn't appear that the authors are plant biologists or use the same words to describe plant biology as plant biologists do. For example, the authors start the ms with some very tired and rejected (e.g., lines 46-47) and with no reference to support the statements. Moreover, the topic of the paper is not going to replace selectable and scorable DNA markers in plant biotech.

We are sorry for the confusion caused by our inaccurate presentation. The main purpose of current study is to establish a RNA-based tool that may complement DNA-based markers for plant biology studies. As pointed out by the other two reviewers, RNA reporter system by itself is a very important tool in functional gene study. Its merit has been abundantly demonstrated in mammalian systems. It was not our intention to even suggest that application of RNA fluorescence aptamer-based reporter will replace DNA markers. In our opinion, they are two different tools that each has its own strength.

We also revised the beginning of manuscript and cited the related reference. Please see lines 33-40 in revised manuscript.

3. The work is largely derivative. The fluorescent aptamer is not really novel in concept and form. Their system is based off the broccoli system and in some places they have not given the correct citation of preceding work; e.g., line 109.

We apologize for not providing the preceding work in previous version, and added the Ref [2] in the revised manuscript. Here, I'd like to point out the novelty in our study compared to previous broccoli system:

(1) The previous broccoli system only used for several high abundance RNA imaging in mammal cells, rarely for genetic coding RNA imaging. Furthermore, in recent years, extensive experiments have been conducted to modify broccoli system, but no modified broccoli has been reported to be applied in plant RNA studies to date, which is largely due to its intrinsically weak fluorescence intensity, low stability and detection limit. In the current study, we made some major structural modification on the original broccoli aptamers and designed a new 3WJ scaffold with a linker. We created a series of aptamers by assembling 3WJ and broccoli in different ways (Fig. S1 and Fig. S2). After screening *in vitro* and *in vivo*, we first obtained the aptamer 3WJ-4×Bro which can be used for RNA imaging in plant cells (seen in Fig. 1, Fig. 2, and Fig. S3-Fig. S7).

(2) We first attempted to introduce 3WJ-4×Bro as a gene marker for reporter system construction, not just for RNA imaging. Meanwhile, we have demonstrated that the new reporter system successfully functioned in both *E. coli* cells and plant cells without perturbation to transcription and translation process (Fig. 4 and Fig. 5).

(3) We applied 3WJ-4×Bro/DFHBI-1T reporter system for transgenic plants screening and identification at transcriptional level, and for the study of plant virus RNA delivery *in vivo* (Fig. 6 and Fig. 7).

4. The experiments and results are insufficient to justify the conclusions. I don't think we're ever told about the spectral characteristics of the system; e.g., optimal excitation, extinction coefficient, etc. The experiments mainly rely on transient expression in benthis, which is ok, but plant biologists will likely be most interested in the system for analysis in stably transgenic plants. Here, the authors recovered transgenic arabidopsis plants, but really very little data are shown and the data shown aren't very thorough or convincing that their system is transformative. Therefore, based on these issues, I am not enthusiastic about this ms.

Thank you for pointing out these critiques. In the revised manuscript, the detailed results have been provided to justify the conclusion sufficiently. In our previous work, we conducted *in vitro* and *in vivo* assays to examine the performance of 3WJ-4×Bro (previous 3WJ-BRO) as a marker. We also unbiasedly enlisted both transient and stable transformation to evaluate the sensitivity and reliability of our reporter system (Fig. 5 and Fig. 7). To make evidence more sufficient, we had enlarged the number of stably transgenic plants and supplemented fluorescence image of T2 transgenic plants, which showed that 3WJ-4×Bro functioned as a stably genetic marker (Fig. 7a). Besides, we also compared 3WJ-4×Bro/DFHBI-1T system with GFP reporter system. The results showed a similar fluorescence intensity between the two systems, suggesting the two systems possess the same sensitivity. Please see Fig. 7c and Fig. 7d in the revised manuscript.

Reviewer #3 (Remarks to the Author):

*This manuscript describes a mRNA detecting marker system for plants. The authors designed a RNA fluorescence aptamer (3WJ-BRO) based on broccoli RNA aptamer. They have shown that the tetrameric version is more sensitive than broccoli aptamer (which is more sensitive than spinach aptamer) for mRNA detection and tracking. Spinach and Broccoli aptamers have been used successfully for imaging of RNA in live bacteria and for tracking the transcription products of RNA pol III in mammalian cells. This type of labeling is not sufficiently sensitive to enable the detection of lower abundance mRNAs. To increase the sensitivity, people have shown to arrange multiple Spinach or Broccoli aptamers in tandem array to increase fluorescence. The authors tagged the 3WJ-BRO on the 3' end of three genes and evaluated the fluorescence in vitro, in DNA-transfected *N. benthamiana* plant protoplasts, in Agro-infiltrated *N. benthamiana* plant leaves, and in transgenic *Arabidopsis* plants. This application with enhanced broccoli RNA-fluorophore pair has not been reported in plant system. RNA reporter system is a very important tool in functional gene study. The use of fluorescent reporter for transcription study in live cells, if sensitive enough, can really positively impact molecular biological research. However, this reviewer is not completely convinced that the fluorescence images reported in this manuscript were the true mRNA localization or simply artifacts of the experimentation.*

We appreciate your thoughtful and supportive comments for our manuscript. As to the concern on whether the fluorescence images are true mRNA localization or simply artifacts, we have supplemented more results to justify that fluorescence signal are true mRNA localization. In short, we simultaneously monitored the red mCherry protein signal and its mRNA with green fluorescence in plant cells expressing *mCherry-3WJ-4×Bro*. We didn't find any cells only showed single green or red fluorescence, which indicated that 3WJ-4×Bro linked with mRNA and did not affect the function of mRNA as genetically encoded template to translate proteins in plant (Fig. 5e). To further demonstrate the green fluorescence only originated from the intact 3WJ-4×Bro tagged mRNA rather than the cleavage products of these mRNA, the cellular total RNAs were extracted from *Nicotiana*

benthamiana leaves expressing 3WJ-4×Bro tagged genes, which showed that a single undegraded band of each RNA sample appeared at the expected position on the gel (Fig. 4d and Fig. S12). The RT-PCR sequencing also suggested a full-length 3WJ-4×Bro tagged mRNA, which illustrated the fluorescence could reflect true mRNA localization. Further, we found the consistency between 3WJ-4×Bro expression and fluorescence emission in transgenic T2 plants (Fig. 7a).

- *Figure 1d, the first and third samples have same label. I think the first sample should be “3WJ-BRO” alone. Please check and correct.*

We corrected the error in the revised version. Thanks for pointing it out.

- *Figure 3, Why are all green fluorescence images showing the localization on cell membranes? Please include images with and without DFHBI-1T for all constructs. What is the size bar length?*

The membrane accumulation of DFHBI-1T from the in vitro experiments was likely due to the actively ongoing endocytosis that mediates the DFHBI-1T intake through membrane fusion. Notably, the membrane accumulation is not the only case in the transient transformed *Nicotiana benthamiana* leaves (Fig. 5c). Besides, we performed an experiment to monitor the dynamic process of mRNA synthesis and transfer in plant cells (Fig. 5e), which showed that green fluorescence also localized in cell cytoplasm and nucleus.

We both had captured all images with and without adding DFHBI-1T for all constructs. No fluorescence was observed in protoplast without DFHBI-1T as expected. Consequently, we used one image with DFHBI-1T as control (Fig. 5a). Here, we supplement the images for all constructs (Fig. S15). The scale bar length is 25 μm and at the top left of the protoplast images.

- *Figure 3, adding DFHBI-1T seems to influence the chloroplast distribution in protoplast, comparing all DFHBI-1T samples with the one sample without DFHBI-1T. It seems that in DFHBI-1T (+) samples, the chloroplasts are moving to the cell membranes.*

Yes, we also noticed that DFHBI-1T seemly has some minor effect on chloroplast distribution in protoplast. But the reason that the chloroplasts seemly moved to the cell membranes is largely due to the tuning of z-axis during imaging by confocal microscope.

- *Figure 3, please show the mCherry expressing protoplasts, and if it overlay image with mCherry-3WJ-BRO.*

The protoplasts assay was initially intended to justify that 3WJ-4×Bro (previous 3WJ-BRO) can be applied for mRNAs imaging in isolated plant cells. To study the effect of 3WJ-4×Bro on translation of fusion mRNA, we have simultaneously showed the mCherry-3WJ-4×Bro RNA and mCherry protein signal in *Nicotiana benthamiana* leaves, which showed part overlay. Please see Fig. 5e in our revised manuscript.

- *Figure 4b is redundant with Figure 2c. The marker lane is empty.*

The previous Figure 2c (now Fig. 4d in revised version) have been used for showing the expression of 3WJ-4×Bro tagged RNAs in *E. coli* cells. While, the previous Figure 4b was used to show the expression of 3WJ-4×Bro tagged RNAs in *Nicotiana benthamiana*, which was removed to supplementary information section (Fig. S12) in revised manuscript. In those assays, total RNAs extracted from cells were loaded and separated in urea-containing agarose gel. The image was taken after the gel was incubated with 10 μM DFHBI-1T at room

temperature for 20 min. Since the marker molecules do not contain aptamer tag and therefore could not bind fluorescent DFHBI-1T, so the marker lane looked empty.

- Figure 4e, need images with plant not Agro-infiltrated.

Following your suggestion, we have added the images from plants not Agro-infiltrated in supplementary information section (Fig. S16).

- Figure 5, need to see mCherry expression images in these lines. All confocal images need to be enlarged to allow readers to view the details.

Thanks for the suggestion. We have provided mCherry expression images and also enlarged images for confocal grafts shown in Fig. 6b (Previous Figure 5).

- Figure 6a, the confocal images should be enlarged to allow readers to view the cellular details. There is no need to show the Arabidopsis plants and their leaf sampling information.

Thanks for your valuable suggestion. We presented the *Arabidopsis* plants and their leaf sampling information to show the growth condition in order to exclude the autofluorescence caused by wound.

- Figure 6b, need to include images from non-transgenic cells.

Thanks for your suggestion. We have added the images of wild type (WT) *Arabidopsis thaliana* as non-transgenic controls (labeled as “WT”) in supplementary information (Fig. S17).

- The manuscript needs to be substantially improved for English and grammar. Sometimes the presentation is so poor, it was very hard to even understand what the authors try to say.

We tried our best to correct the grammar mistakes and inaccurate usage in the revised version. In addition, to bring to the reviewers a more readable manuscript, we also invited a native speaker to improve the English of the paper.

Yun Zhao, PhD

Professor in Plant Genetics

School of Life Sciences

Sichuan University

Address: No.29 Wangjiang Road, Chengdu, Sichuan, China, 610064

Phone: 86-028-85412485

E-mail: zhaoyun@scu.edu.cn

Reviewers' Comments:

Reviewer #2:

Remarks to the Author:

The writing is still not publication-quality, and it appears that the introductory text (and other text) that is quite old-sounding and not relevant to plant biology remains. Indeed, few of my critical comments were adequately addressed in the revision.

Reviewer #4:

Remarks to the Author:

The manuscript has been well written, and most of the comments have been well addressed with good statistical analysis. Enough experimental details have been provided. It is interesting though "Reif R et al. Fluorogenic RNA Nanoparticles for Monitoring RNA Folding and Degradation in Real Time in Living Cells. *Nucleic Acid Ther.* 2012 22(6):428-37" seems not be cited anywhere in the paper.

Reviewer #5:

None

Detailed Response to the Reviewers.

Reviewer #2 (Remarks to the Author):

The writing is still not publication-quality, and it appears that the introductory text (and other text) that is quite old-sounding and not relevant to plant biology remains. Indeed, few of my critical comments were adequately addressed in the revision.

Following the reviewer's suggestion, we have restructured the introduction part and added more references to recent progress in the field. To address the other concern regarding the writing quality, we have also enlisted professional assistance from the Plant Biology Science Editors Service. We hope that this effort helps us to further improve our manuscript's readability.

To address the reviewer's previous main concerns about the spectral characteristics and the plant stable transformation, we have included a few new experimental data. Particularly, the optimal excitation is now included as Fig 1a and extinction coefficient as Table 1. In addition, we have performed a detailed genetic analysis of the stably transformed T1 and T2 plants. In total, we screened 6000 T₁ seeds and analyzed 766 T₂ plants from 6 transgenic families. The data is now included as Fig. 7a, b and Table 2. These analyses have verified again that 3WJ-4×Bro/DFHBI-1T system is genetically stable and reliable.

Reviewer #4 (Remarks to the Author):

The manuscript has been well written, and most of the comments have been well addressed with good statistical analysis. Enough experimental details have been provided. It is interesting though "Reif R et al. Fluorogenic RNA Nanoparticles for Monitoring RNA Folding and Degradation in Real Time in Living Cells. Nucleic Acid Ther. 2012 22(6):428-37" seems not be cited anywhere in the paper.

We greatly appreciate the reviewer's supportive comments. The study mentioned is indeed very relevant to ours. We have now included it into our reference, listed Ref. [20].

Reviewer #5:

1. Fig.4D and Fig.S12 are still missing marker labeling. Although the marker bands are not visible, the authors should mark the band locations on the membrane.

Thanks for the suggestion. We have added marker labeling in the revised in Fig. 4d and Fig. S14 (previously Fig. S12)

2. Fi.3b label may not be correct because the -DFHBI-1T and +DFHBI-1T may be mistakenly switched

We apologize for the error. We have corrected it in the current manuscript

3. Fig. S8 contains misspelling of "mCheery" which could be "mCherry" for the gene in the middle construct.

We have corrected the misspelling in the revised manuscript.

4 Fig.S17 is missing.

We are sorry for the error. We have now added the previous Fig. S17 into Fig. 6c; this is now the left part of Fig. 6c, marked as “Wild Type”.

5. There are misspelling the manuscript text.

Thanks for pointing it out. We have double-checked the spelling in the revised manuscript.

6 The Written English still needs improving to meet needs of the journal quality.

Following the reviewer’s suggestion, we have enlisted professional editing assistance from the Plant Biology Science Editors Service to improve language of the revised manuscript. We hope that this effort improved the readability.

Reviewers' Comments:

Reviewer #5:

Remarks to the Author:

This revised manuscript version has much improvements by addressing all concerns. At the same time, there are still minor issues to be fixed before it can be published. These minor issues are as follows:

The word "transformative" in "Plant genetic transformative reporter systems" is a wrong wording. The correct word should be "transformation" so that the correct wording should be "Plant genetic transformation reporter systems" throughout the manuscript.

The word "descendant" in the wording "descendant generations" is used too frequently so the "progeny generations" should be used from place to place in the manuscript because many researchers prefer to using progeny instead of descendant so they can connect the terms seamlessly.

Line 38, the word "isolate" is a wrong choice because of inaccuracy. The correct word should be "enrich" because you really enrich, but not isolate, the transgenic cells.

Line 53, please add "as" before "its detection requires" so that "as its detection requires" will read better.

Lines 83 and 520, the square signs need to be removed. Please check out remaining manuscript to remove any additional square signs.

In the Discussions, authors should propose or speculate the reason for 3WJ-4×Bro to be the brightest and most stable as compared with other RNA-aptamers.

In Methods, the Arabidopsis floral dip protocol (Clough and Bent 1998?) followed by the authors should be cited. The application time and concentration of DFHBI-1T to assay stably-transformed Arabidopsis plants should be provided.

Figure legends: the letter "t" in the term "t-text" should be italic throughout the text and figure legends.

For each complex figure, the statement "Source data are provided as a Source Data Underlying Fig. XX" can be used only once at the end of figure legends instead of being used repeatedly for each sub figure. For example, Source data are provided as Source Data Underlying Fig. 5a-5e, respectively. This is to make the presentation more concise by avoiding redundancy.

Zhanyuan J. Zhang

Dear Reviewer

Thank you so much for your help and comments to improve our manuscript. After carefully reading the comments, we have revised our manuscript in detail. The point-to-point responses are also provided here.

Detailed Response to the Reviewers.

1. The word “transformative” in “Plant genetic transformative reporter systems” is a wrong wording. The correct word should be “transformation” so that the correct wording should be “Plant genetic transformation reporter systems” throughout the manuscript.

Thanks for your help. We have changed “Plant genetic transformative reporter systems” as “Plant genetic transformation reporter systems” throughout the manuscript.

2. The word “descendant” in the wording “descendant generations” is used too frequently so the “progeny generations” should be used from place to place in the manuscript because many researchers prefer to using progeny instead of descendant so they can connect the terms seamlessly.

We have replaced the “descendant generations” with “progeny generations” in revised manuscript.

3. Line 38, the word “isolate” is a wrong choice because of inaccuracy. The correct word should be “enrich” because you really enrich, but not isolate, the transgenic cells.

Thanks for pointing it out, we have changed the word “isolate” as “enrich”

4. Line 53, please add “as” before “its detection requires” so that “as its detection requires” will read better.

Thanks for the advice. We have added “as” before “its detection requires”

5. Lines 83 and 520, the square signs need to be removed. Please check out remaining manuscript to remove any additional square signs.

Sorry, we found that the square signs are some symbols, such as roman numerals II, which are misidentified by PDF version converted from Microsoft Word versions in the resubmission system. We have provided a compatible Microsoft Word version for submission.

6. In the Discussions, authors should propose or speculate the reason for 3WJ-4×Bro to be the brightest and most stable as compared with other RNA-aptamers.

We have added the discussion on the reason for 3WJ-4×Bro to be the brightest and most stable as compared with other RNA-aptamers. Pleas see line 359-363, marked by red font.

7. In Methods, the Arabidopsis floral dip protocol (Clough and Bent 1998) followed by the authors should be cited. The application time and concentration of DFHBI-IT to assay stably-transformed Arabidopsis plants should be provided.

Arabidopsis floral dip protocol (Clough and Bent 1998) have been cited in method section, listed Ref. [47], marker by red font. In stably-transformed Arabidopsis plants assay, 30 μ M DFHBI-1T were infiltrated into plant and incubated for 30 min. please see lines 585-589, marked by red font

8. Figure legends: the letter “t” in the term “t-text” should be italic throughout the text and figure legends

Thanks for your comment, and we have changed the letter “t” in the term “t-text” as italic form throughout the text and figure legends

9 For each complex figure, the statement “Source data are provided as a Source Data Underlying Fig. XX” can be used only once at the end of figure legends instead of being used repeatedly for each sub figure. For example, Source data are provided as Source Data Underlying Fig. 5a-5e, respectively. This is to make the presentation more concise by avoiding redundancy.

Thank you for the advice, we have changed the statement of Source data, which is presented only once at the end of figure legends and marked by red font.